# The E2F1-HMGCR axis promotes ferroptosis resistance in immune refractory tumor cells

Sung Wook Son[1,11], Hyo-Jung Lee[2,3,4,11], NaNa Kang[1], Seongjae Bae[1], Eunho Cho[2,3], Heeju Kwon[2,3,4], Da-Young Yoon[2,3,4], Chaeleen Lee[2,3,4], Seungho Lee[2,3,4], Min Kyu Son[2,3,4], Jisu Chae[2,3,4], Suyeon Kim[2,3], Se Jin Oh[2,3,4], Younji Sim[2,3], Kyung-Mi Lee[2,3,4], Cassian Yee[5], Seon Rang Woo[1], Yun-Jeong Jeong[1], Hee-Jung Choi[1,6], Jong-Young Kwak[7], Eun-Woo Lee[8], Jinuk Park[1,9], Sang Gyu Kwak[10], Young-Chae Chang[1] ✉, Tae Woo Kim[2,3,4] ✉ & Kwon-Ho Song[1] ✉

During cancer immunoediting, cancer cells deregulate cell death executioner mechanisms to escape immunotherapy-induced antitumor immunity. Ferroptosis, a type of regulated necrosis triggered by lipid peroxidation, plays a pivotal role in the anti-tumor activity of T cell-based immunotherapies; however, mechanisms for the modulation of ferroptosis in immune-refractory tumor cells are unclear. In this study, using preclinical models of immune refractory tumors obtained following the course of immunoediting by PD-1 blockade and adoptive T cell therapy (ACT), we find that T cell-based immunotherapy drives the development of ferroptosis resistance of tumor cells. In this process, E2F1 is upregulated by immunotherapy and it in turn binds to the promoter of the HMGCR gene to upregulate HMGCR, thereby contributing to the resistance to ferroptosis. Notably, HMGCR inhibition renders immune-refractory tumor cells susceptible to ACT and PD-1 blockade. Thus, our results reveal a mechanism by which cancer cells modulate ferroptosis to acquire resistance to immunotherapy and implicate the E2F1-HMGCR axis as a central molecular target for controlling ferroptosis resistance of immune-refractory cancer.

Cancer immunotherapy using immune checkpoint blockade (ICB) has shown remarkable efficacies in the treatment of multiple cancer types[1]. However, the presence of immune-refractory tumor cells blocking amplification of anti-tumor immunity limits its clinical success[2,3]. An expanding body of evidence suggests that cancer immunoediting drives the adaptation of tumor cells to host immune surveillance, contributing to the generation of tumor cells with survival advantages[4–7]. In this regard, we have demonstrated that immune selective pressures imposed by anti-PD-1 therapy and tumor antigen (TA)-specific cytotoxic T lymphocytes (CTLs) facilitate the

[1]Department of Cell biology, Daegu Catholic University School of Medicine, Daegu, South Korea. [2]Department of Biochemistry and Molecular Biology, Korea university College of Medicine, Seoul, South Korea. [3]Department of Convergence Medicine, College of Medicine, Korea University, Seoul, South Korea. [4]BK21 Graduate Program, Department of Biomedical Science, Korea university College of Medicine, Seoul, South Korea. [5]Department of Melanoma Medical Oncology and Immunology, U.T. MD Anderson Cancer Center, Houston, TX, USA. [6]Jin BioCell Co. Ltd. R&D Center, Yangsan, South Korea. [7]Department of Pharmacology, School of Medicine, Ajou University, Suwon, South Korea. [8]Metabolic Regulation Research Center, Korea Research Institute of Bioscience and Biotechnology (KRIBB), Daejeon, South Korea. [9]Department of Medicine, Daegu Catholic University School of Medicine, Daegu, South Korea. [10]Department of Medical Statistics, Daegu Catholic University School of Medicine, Daegu, South Korea. [11]These authors contributed equally: Sung Wook Son, Hyo-Jung Lee. ✉e-mail: ycchang@cu.ac.kr; twkim0421@korea.ac.kr; khsong@cu.ac.kr

enrichment of a subset of tumor cells with resistance to CTL-mediated killing[5,8]. Given the crucial role of T cell-mediated killing of tumor cells in initiating or reinvigorating the cancer-immunity cycle through the presentation of TAs to CTLs[9–11], these results suggest the intrinsic resistance to CTLs of immune-edited tumor cells as a critical obstacle to improving immunotherapy outcome. Therefore, understanding the mechanisms through which immunoediting promotes the resistance of tumor cells to CTLs will provide valuable insights for the development of strategies to further enhance the anti-tumor immunity.

It has been well documented that CTLs activated by cancer immunotherapy execute tumor clearance mainly by inducing apoptosis through the granule exocytosis pathway and the Fas death receptor pathway[12,13]. Nevertheless, recent studies have highlighted that CTL-induced ferroptosis plays an essential role in cancer immunotherapy by inducing immunogenic cell death and eliciting anti-tumor immune responses[14]. Ferroptosis is a type of regulated cell death characterized by iron-dependent accumulation of phospholipid peroxides[15], it is regulated by a complex interplay among iron, lipid and cysteine metabolisms[16]. Ferroptotic cell death can be triggered through either the extrinsic or the intrinsic pathway. The extrinsic pathway is initiated through inhibiting the cystine-glutamate antiporter (system xc⁻) or activating iron transporters, such as transferrin and lactotransferrin, whereas the intrinsic pathway is mainly induced by blocking the expression or activity of intracellular antioxidant enzymes, such as glutathione peroxidase 4 (GPX4)[17–19]. Although ferroptosis can both promote and inhibit tumor development and progression, it is typically considered as a tumor suppressor[20]. Therefore, targeting ferroptosis may reveal new therapeutic opportunities in cancer. Indeed, it has been suggested that ferroptosis induction in combination with immune checkpoint inhibition or radiation is promising in cancer therapy[21]. However, varying susceptibilities to ferroptosis have been observed among different cancer cell types, due to mechanisms such as altered lipid metabolism and anti-oxidative defenses[22–24].

The 3-hydroxy-3-methylglutaryl-CoA reductase (HMGCR) is a rate-limiting enzyme in the mevalonate pathway for cholesterol biosynthesis[25]. Interestingly, it has been well-documented that products of the HMGCR-mediated mevalonate pathway, such as isopentenyl pyrophosphate (IPP) and the antioxidant CoQ10, protect against ferroptosis in several types of cancer[26]. Notably, inhibiting HMGCR with statins downregulates the mevalonate pathway and results in the induction of ferroptosis[27,28]. In addition, several preclinical and retrospective clinical studies have shown that statins could enhance the efficacy of T cell-based immunotherapy[29,30]. Despite growing importance of HMGCR as a therapeutic target, the potential link between HMGCR and cancer immunoediting remains poorly understood.

Here, we report the mechanism by which immune editing promotes resistance to ferroptosis in tumor cells. In this process, E2F1 upregulates HMGCR through binding to its promoter, thereby diminishing the oxidative damage induced by immunotherapy. Importantly, HMGCR inhibition reverses ferroptosis resistance of immune-edited tumor cells and increases their susceptibility to adoptive T cell transfer as well as PD-1 blockade. Thus, our data provide evidence that the E2F1-HMGCR axis plays a central role in controlling the ferroptosis resistance of immune-refractory cancer.

## Results

### HMGCR upregulation following ICB-mediated immune selection promotes ferroptosis resistance

To understand the mechanisms by which ICB-mediated immune editing promotes the intrinsic resistance to CTLs in tumor cells, we employed two anti-PD-1 therapy-refractory tumor models. B16 P3

and TC-1 P3 were generated from ICB-susceptible mouse melanoma cell line B16 (B16 P0)[31] and mouse lung cancer cell line TC-1 (TC-1 P0) which expresses the E7 oncoprotein from human papillomavirus 16[32], respectively, through three rounds of in vivo selection by anti-PD-1 therapy (Supplementary Fig. 1a). Notably, relative to ICB-susceptible respective parental (P0) cells, the P3 cells were resistant to anti-PD-1 treatment in vivo and were refractory to apoptotic death by cognate CTLs[5] (Supplementary Fig. 1b, c). However, there was no significant difference in the level of MHC class I or tumor antigens between P0 and P3 cells[5] (Supplementary Fig. 1d, e). These data suggested that the CTL-refractory property of both B16 P3 and TC-1 P3 tumor cells over the course of anti-PD-1 therapy-mediated immune editing might be originated from resistance to CTL killing rather than T cell recognition.

Given ferroptosis, besides apoptosis, being a key anti-tumor mechanism promoted by CTL during ICB therapy[14], we asked whether tumor cells acquire resistance to ferroptosis through ICB-mediated immune editing. In this regard, activated CTL-derived supernatants have been reported to promote both apoptosis and ferroptosis[14,33,34]. To test this, we treated P0 and P3 cells with CTL-derived supernatants and assessed ferroptosis (characterized by lipid peroxidation) and apoptosis (marked by active caspase-3). Notably, CTL-derived supernatants increased the frequency of active caspase-3⁺ cells and elevated lipid ROS levels in immune-susceptible P0 cells (Supplementary Fig. 2a, b). These effects were partially blocked by the apoptosis inhibitor zVAD or the ferroptosis inhibitor Lip-1 (Supplementary Fig. 2a, b), indicating that both apoptosis and ferroptosis contribute to CTL-mediated tumor cell killing. In contrast, immune-refractory P3 cells exhibited resistance to both forms of cell death (Supplementary Fig. 2a, b). In addition, we compared the susceptibility of P3 versus P0 cells to RSL3 and erastin, known ferroptosis inducers that work through inhibiting GPX4 and cystine/glutamate antiporter (xCT), respectively[19,35]. Based on the IC$_{50}$ values, indicating the concentration of drugs that inhibits 50% of cell viability, we found that P3 cells were less sensitive to the ferroptosis inducers than P0 cells (Fig. 1a, b). Indeed, RSL3-induced death in P0 cells was accompanied by increased lipid ROS production, both of which were reversed by Lip-1 but not by zVAD (Fig. 1c, d), indicating that RSL3 selectively induces ferroptosis in P0 cells. However, P3 cells did not show a significant difference in the cell death or the production of lipid ROS upon treatment with RSL3 (Fig. 1c, d), indicating ferroptosis resistance. These findings suggest that anti-PD-1 therapy-mediated immune selection promotes tumor cell resistance to ferroptosis in response to cytotoxic T cell attacks.

It has been documented that ferroptosis is regulated by key factors involved in iron, lipid and antioxidant metabolisms[16]. To explore the mechanism responsible for the resistance of tumor cells to ferroptosis, we compared the expression of genes involved in the negative regulation of ferroptosis between P0 and P3 cells of B16 and TC-1 (Fig. 1e). Among the 17 genes responsible for the negative regulation of ferroptosis[16,36], we noted that the HMGCR gene was upregulated in both B16 and TC-1 P3 cells compared to their P0 cells (Fig. 1f). Indeed, P3 cells had higher HMGCR protein level and activity compared with P0 cells (Fig. 1g, h), without changes of the protein levels and activity of GPX4 or the protein levels of SLC7A11 (Supplementary Fig 3a, b), which are well-known regulators of ferroptosis resistance[37,38]. To directly link HMGCR up-regulation to ferroptosis resistance, we silenced HMGCR in B16 P3 and TC-1 P3 cells (Supplementary Fig 4a, b) and observed increased sensitivity of P3 cells to ferroptotic cell death induced by RSL3, which was accompanied by increased levels of lipid ROS (Fig. 1i, j and Supplementary Fig 5), regardless of GPX4 activity. Therefore, we concluded that HMGCR was upregulated as tumor cells undergo ICB-mediated immune selection and contributed to the ferroptosis resistance of tumor cells.

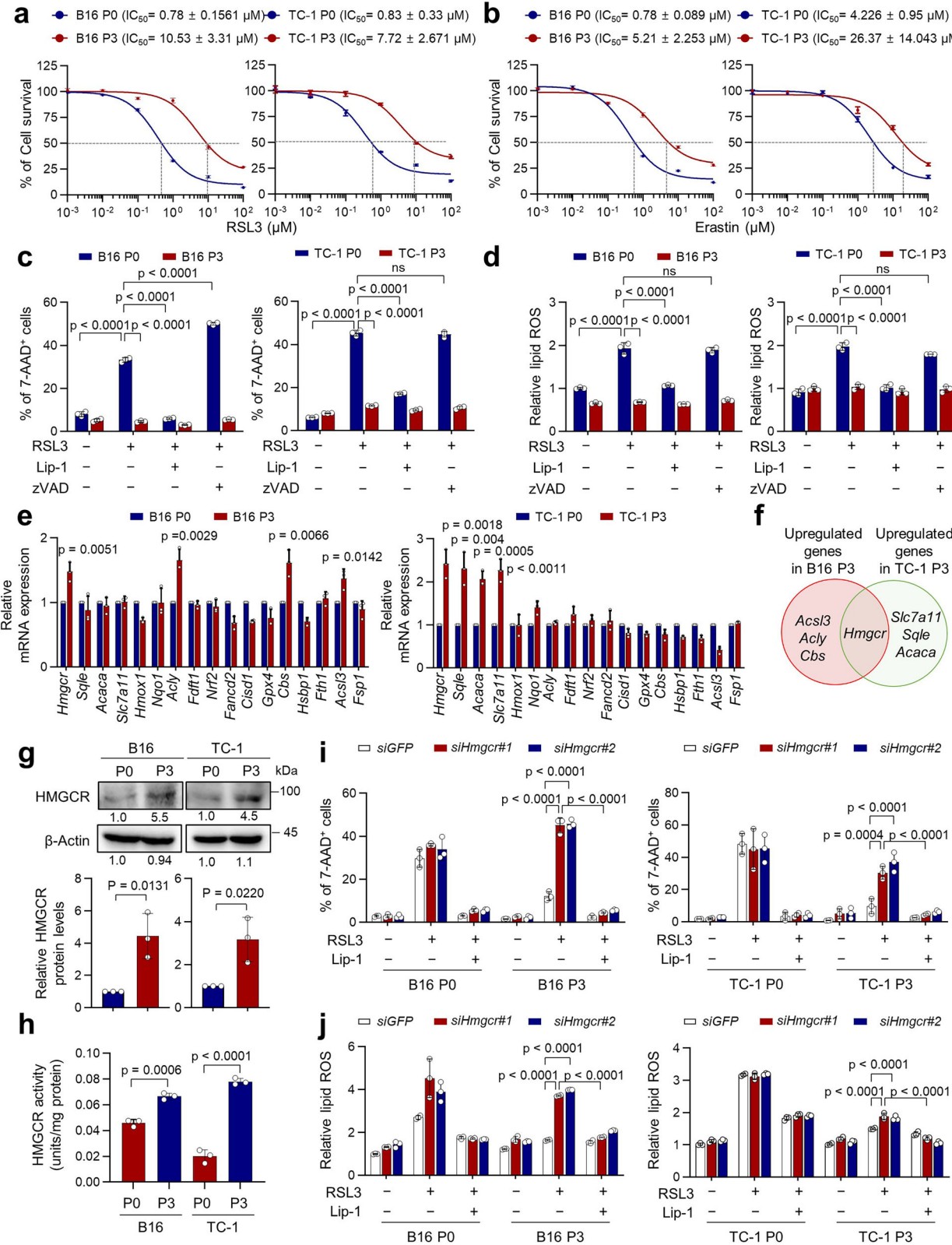

## Loss of HMGCR in ICB-refractory tumor cells enhances ICB-mediated anti-tumor immunity by promoting ferroptosis

Since ferroptosis is found to play essential roles in anti-tumor immunity and in regulating the immune response to ICB[14], we reasoned that silencing of HMGCR expression could reverse the resistance of ICB-refractory tumor to anti-PD-1 therapy. We treated B16 P3 tumor-bearing mice with an anti-PD-1 antibody along with chitosan nanoparticles (CNPs) carrying *siHmgcr* or *siGFP* (Fig. 2a and Supplementary Fig. 6). While the anti-PD-1 antibody alone did not affect the tumor growth, its combination with *siHmgcr #1 or #2*-CNPs profoundly retarded the tumor growth, which was reversed by the simultaneous Lip-1 treatment (Fig. 2b and Supplementary Fig. 6). Furthermore, while PD-1 blockade alone failed to increase the percentage of 7AAD+ tumor cells and

**Fig. 1 | HMGCR upregulation following ICB-mediated immune selection promotes ferroptosis resistance.** B16 P0, P3 cells and TC-1 P0, P3 cells were treated with indicated concentrations of RSL3 (**a**) or Erastin (**b**) for 24 h. Cell viability was measured by Trypan blue exclusion assay. The concentrations causing 50% inhibition of cell viability ($IC_{50}$ values) were determined. **c, d** B16 P0, P3 cells and TC-1 P0, P3 cells were treated with RSL3 (1 or 2 µM) in the absence or presence of zVAD (10 µM) or Lip-1 (1 µM) for an additional 20 h. The percentage of 7-AAD+ cells **c** and relative lipid ROS **d** were measured by flow cytometry. The data are representative of those from 3 independent experiments with triplicate. **e** The real time PCR array analysis for ferroptosis negative regulator genes in B16 P0, P3 and TC-1 P0, P3. **f** Venn diagram showing the overlap of ferroptosis negative regulator genes

between upregulated in B16 P3 versus P0 (red) and those upregulated in TC-1 P3 versus P0 (green). **g** HMGCR protein levels in B16 P0, B16 P3, TC-1 P0, and TC-1 P3 was determined by Western blot. β-actin was included as an internal loading control. **h** The HMG-CoA reductase activity in B16 P0, B16 P3, TC-1 P0, and TC-1 P3 ($n = 3$; in duplicate). Flow cytometry analysis of the 7AAD+ cells (**i**) and relative lipid ROS (**j**) in indicated cells treated with RSL3 with or without Lip-1 (1 µM) for 20 h. All data are representative of those from 3 independent experiments with triplicate. The error bars represent mean ± SD. All $p$ value are presented exactly in figure. The $p$ values by two-way ANOVA (**c, d, i, j**), and unpaired, two-tailed Student's $t$ test (**e, g, h**) are indicated. Source data are provided as a Source data file.

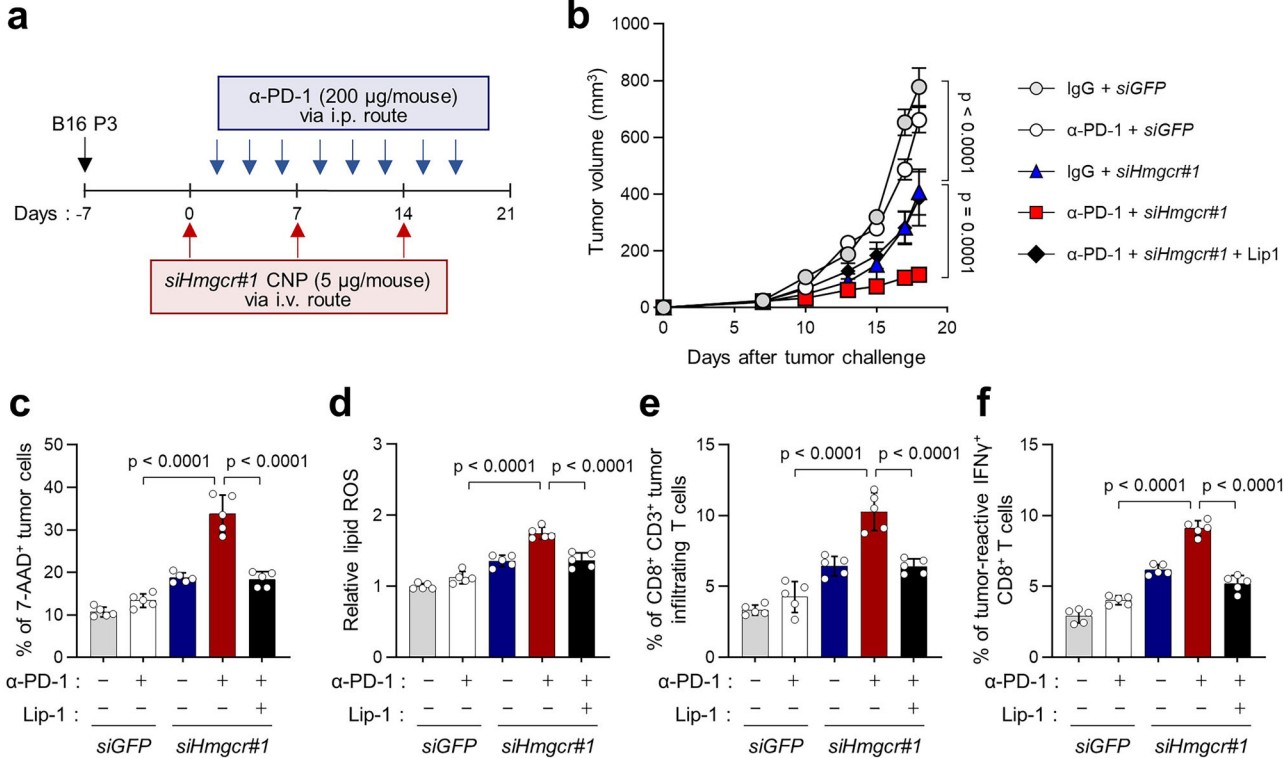

**Fig. 2 | HMGCR depletion reverses immune-refractory features by inducing the anti-PD-1-mediated ferroptotic cell death. a** Schematic of the therapy regimen in mice implanted with B16 P3 cells. **b** Tumor growth during 18 days after mice inoculated with B16 P3 then treated with the indicated reagents. The percentage of 7-AAD+ cells (**c**) and relative lipid ROS (**d**) measured by flow cytometry. B16 P3 tumor-bearing mice were administered *siGFP* or *siHmgcr#1* with anti-PD-1 or anti-

PD-1 plus Lip-1 as indicated. **e** Flow cytometric profiles of tumor-infiltrating CD3+ CD8+ T cells. **f** Percentage of IFNγ+ to tumor-infiltrating CD3+ CD8+ T cells. For the in vivo experiments, 7 mice from each group were used, and randomly selected 5 samples were analyzed. The error bars represent mean ± SD. All $p$ value are presented exactly in figure. The $p$ values by two-way ANOVA (**b**) and one-way ANOVA (**c–f**) are indicated. Source data are provided as a Source data file.

the lipid ROS level, the combined treatment of anti-PD-1 antibody and *siHmgcr*-CNPs significantly increased the percentage of 7AAD+ tumor cells and the lipid ROS level, both of which were reversed by the concomitant Lip-1 treatment (Fig. 2c, d). These data indicated that HMGCR depletion could reverse the ICB refractory phenotype of B16 P3 tumor by promoting ferroptosis.

It has been demonstrated that immunogenic cell death such as ferroptosis and necroptosis under anti-tumor therapy could lead to the release of TAs that prime TA-reactive T cells and reinitiate subsequent positive feedback loop of anti-tumor immune responses (known as the cancer-immunity cycle)[39]. To verify whether the generation of tumor-reactive T cells was affected by the HMGCR depletion in ICB-refractory B16 tumor, we further analyzed the CD8+ T cells in tumor. Notably, the percentage of CD8+ T cells infiltrating the tumor was significantly increased in those with the combined treatment of

anti-PD-1 antibody and *siHmgcr*-CNPs, compared to those with either treatment alone (Fig. 2e). Furthermore, to confirm tumor antigen-specific T cell responses rather than nonspecific T cells activation, we stimulated tumor-derived single cells with either DMSO or gp100, a B16 tumor-associated antigen, then assessed IFNγ-producing CD8+ T cells. We found that the combination treatment group had increased tumor-reactive IFNγ+ CD8+ T cells compared to other treatment groups (Fig. 2f). Importantly, increased levels of tumor infiltrating CD8+ T cell and tumor-reactive IFNγ+ CD8+ T cells by the combination treatment of anti-PD-1 and *siHmgcr*-CNPs were reversed upon Lip-1 treatment (Fig. 2e, f). Taken together, our data demonstrated that targeting HMGCR could induce tumor-reactive T cell response via tumor ferroptosis, thereby enhancing the therapeutic efficacy of anti-PD-1 therapy. Thus, these findings suggest that increased expression of HMGCR in ICB-refractory tumor cells may influence immune refractory features of the tumor microenvironment (TME), such as limited

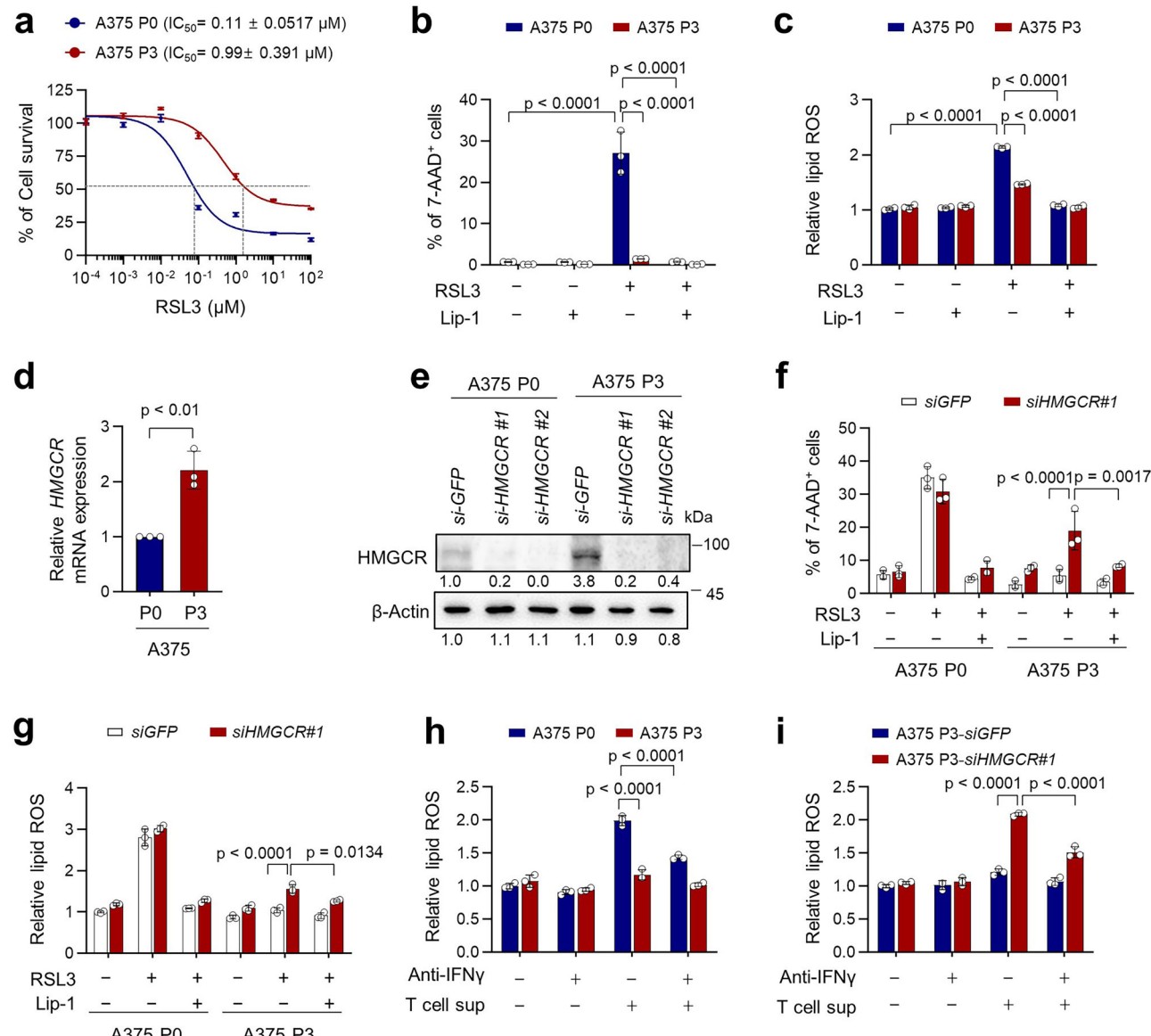

**Fig. 3 | CTL-mediated immune selection triggers ferroptosis-resistance by upregulating HMGCR. a** A375 P0 and P3 cells were treated with indicated concentrations of RSL3 for 24 h. Cell viability was measured by Trypan blue exclusion assay and the concentrations of RSL3 causing 50% inhibition of cell viability (IC$_{50}$ values) were determined. A375 P0 and P3 cells were treated with RSL3 (200 nM) in the absence or presence of Lip-1 (1 μM) for an additional 20 h, and then the percentage of 7-AAD$^+$ cells (**b**) and relative lipid ROS (**c**) were measured by flow cytometry. **d** The HMGCR mRNA levels of immune-resistance A375 cells were determined by qRT-PCR. **e** The levels of HMGCR protein in A375 P0 and P3 cells transfected with siRNA targeting *GFP* or *HMGCR* were measured by Western blot. Flow cytometry analysis of the 7AAD$^+$ cells (**f**) and relative lipid ROS (**g**) in indicated cells treated with RSL3 (200 nM) with or without Lip-1 (1 μM) for 24 h. **h, i** Tumor cells were incubated with the supernatant from tumor-specific CTLs with or without anti-IFNγ. The lipid ROS was measured by flow cytometry. All in vitro experiments were performed in triplicate. The error bars represent mean ± SD. All p value are presented in figure. The *p* values by two-way ANOVA (**b, c, f, g, h, i**), and unpaired, two-tailed Student's *t* test (**d**) are indicated. Source data are provided as a Source data file.

infiltration of T cells into the tumor and resistance of tumor cells to CTLs.

## CTL-mediated selection evolves ferroptosis-resistance by upregulating HMGCR

It was unclear whether HMGCR upregulation leading to ferroptosis resistance in ICB-refractory tumor cells was a consequence of immune selection mediated by TA-specific CTLs or a consequence of the particular ICB system that we used. Since TA-specific CTLs are key components in ICB-mediated anti-tumor effects, we postulated that HMGCR upregulation following ICB-mediated immune selection was due to selection imposed by CTLs. To test this, we employed A375

human melanoma cells as a representative for the clinical application of adoptive CD8$^+$ T cell transfer therapy (ACT). We established an ACT-refractory A375 P3 model from the parental A375 cells (A375 P0) through three rounds of in vivo selection by HL-A2-restricted NYESO1-specific CD8$^+$ T clone (MAK #11)[40]. Intriguingly, like the ICB-refractory mouse tumor models, CTL-refractory human A375 P3 cells were more resistant to ferroptosis induced by either CTL-derived supernatant or RSL3 treatment than A375 P0 cells (Fig. 3a and Supplementary Fig. 7a, b). Indeed, RSL3 effectively induced cell death of A375 P0 cells, which was accompanied by increased production of lipid ROS, and these effects could be reversed upon Lip-1 treatment whereas A375 P3 cells did not show a significant difference in cell death or production of

lipid ROS upon treatment with RSL3 (Fig. 3b, c). Moreover, we noted that levels of HMGCR mRNA and protein were upregulated in A375 P3 cells compared to A375 P0 cells (Fig. 3d, e), without changes of the expression of GPX4 and SLC7A11 (Supplementary Fig. 8a, b). Silencing HMGCR in A375 P3 cells increased sensitivity to ferroptotic cell death induced by RSL3, which was accompanied by increased levels of lipid ROS (Fig. 3f, g). Given the crucial role of HMGCR upregulation in the ferroptosis-resistant phenotype of P3 cells, we examined the effect of HMGCR overexpression in P0 cells. Notably, HMGCR overexpression alone was sufficient to confer resistance to RSL3-induced ferroptosis, as indicated by a reduced proportion of 7-AAD+ cells and decreased lipid ROS levels (Supplementary Fig. 9a–c). Together, these data indicate that CTL-mediated immune selection drives ferroptosis resistance through upregulation of HMGCR.

Previously, it has been reported that activated T cell-derived IFNγ triggers ferroptosis, thereby contributing to the anti-tumor efficacy of immunotherapy[14]. In this regard, we observed that supernatant from activated-NY-ESO1-specific T cells could increase cell death and lipid ROS in A375 P0 cells and that these effects were abolished by IFNγ neutralizing antibodies or Lip-1 (Fig. 3h and Supplementary Fig. 10a, b), indicating a crucial role of soluble IFNγ in the ferroptotic cell death induced by CTLs. Consistent with our results from RSL treatment, supernatant from the activated CTLs alone failed to significantly increase cell death or lipid ROS in A375 P3 cells (Fig. 3h and Supplementary Fig. 10a, b), however these refractory phenotypes of A375 P3 cells were reversed after HMGCR silencing (Fig. 3i and Supplementary Fig. 10c, d). Notably, the increase in lipid ROS induced by the supernatant from *siHMGCR*-treated A375 P3 could be abolished by IFNγ neutralizing antibodies or Lip-1 (Fig. 3i and Supplementary Fig. 10c, d). These results suggest that HMGCR upregulation in tumor cells over the course of CTLs-mediated immune editing could promote resistance to ferroptosis mainly induced by activated T cell-derived IFNγ.

### E2F1 transcriptionally activates the expression of HMGCR to induce ferroptosis resistance

We next attempted to elucidate the mechanism responsible for HMGCR up-regulation in immune-edited tumor cells. Although it has been well documented that SREBPs are the key regulators of HMGCR gene[41], there were no significant increase in the expression of SREBP1 and 2 genes in CTL-refractory P3 tumor cells and NR to anti-PD-1 therapy compared to P0 tumor cells and R to anti-PD-1 therapy, respectively (Supplementary Fig 11). Instead, we found that E2F1 among the 33 transcription factors that might be responsible for the transcriptional regulation of the HMGCR gene, identified from the CHEA Transcription Factor Binding Site Profiles dataset (https://maayanlab.cloud/Harmonizome/gene/HMGCR), was upregulated in both non-responders (NR) to anti-PD1 therapy and P3 cells compared with responders (R) and corresponding P0 cells, respectively (Fig. 4a). Interestingly, we identified two highly conserved putative E2F1-binding elements in the promoter regions of both human and mouse HMGCR genes, supporting the possibility that E2F1 functions as a direct transcriptional activator of HMGCR (Fig. 4b). Indeed, silencing of E2F1 in A375 P3 cells decreased both HMGCR protein and mRNA levels (Fig. 4c, d). We then engineered a reporter that expressed luciferase under the control of the human HMGCR promoter (pGL3-HMGCR pro) (Fig. 4e). Notably, the promoter activity was higher in A375 P3 cells than in A375 P0 cells; however, the increased promoter activity in A375 P3 cells was diminished by E2F1 depletion (Fig. 4f). In addition, mutation of either or both E2F1-binding sites in the HMGCR promoter decreased the luciferase activity in A375 P3 cells, suggesting that the E2F1-binding sites might acts as cis-acting elements responsible for the basal activity of HMGCR promoter (Fig. 4e, g). Quantitative chromatin immunoprecipitation (qChIP) assays using primers targeting the two adjacent E2F1-binding sites confirmed the direct binding of E2F1 to the putative

binding region within the HMGCR promoter and showed that E2F1 binding was greater in P3 cells than in P0 cells (Fig. 4h). These findings demonstrate that E2F1 up-regulates HMGCR transcription by directly binding to its promoter region.

Given the role of E2F1 as an upstream regulator of HMGCR, we reasoned that E2F1 might be responsible for ferroptosis resistance of immune refractory tumor cells. Indeed, E2F1 knockdown in both A375 and TC-1 P3 cells re-sensitized them to RSL3 treatment, accompanied by increased lipid ROS production that could be inhibited by Lip-1 treatment (Fig. 4i, j and Supplementary Fig. 12). However, *siE2F1* treatment did not significantly alter the susceptibility of P0 cells to RSL3 (Fig. 4i, j and Supplementary Fig. 12). To complement above results from reducing E2F1 in P3 cells, we then investigated whether E2F1 over-expression in P0 cells could promote ferroptosis-resistance. Ectopic expression of E2F1 in A375 P0 cells increased both HMGCR mRNA and protein levels (Fig. 4k and Supplementary Fig. 13) as well as reduced ferroptotic cell death and lipid ROS production induced by RSL3 (Fig. 4l, m). Notably, HMGCR knockdown in E2F1-overexpressing A375 P0 cells increased their susceptibility to ferroptosis induced by RSL3, which was accompanied by increased lipid ROS production (Fig. 4l, m). Thus, our findings demonstrate that E2F1 is a key mediator that determines resistance to ferroptosis through regulation of HMGCR expression.

We next investigated the underlying mechanism responsible for the upregulation of E2F1 during immune editing. Our previous study showed that CTL-mediated immune selection enriches immune-refractory cancer cells with cancer stem cell (CSC)-like properties[9,42]. In this regard, E2F1 contributes to CSC-like properties by activating FGFR signaling or inducing the stemness factor NANOG, and is further amplified through a reciprocal regulatory loop in which NANOG-driven AKT hyperactivation impairs RB function, thereby enhancing E2F1 activity[42,43]. Based on this, we hypothesized that E2F1 upregulation may result from immune selection pressure induced by ICB or ACT therapies. To investigate this, we evaluated E2F1 expression in TC-1 and A375 cells across successive rounds of immune selection (P0 to P3) using ICB and ACT, respectively. We observed a stepwise increase in both E2F1 expression and proportion of E2F1+ cells from P0 to P3 cells (Supplementary Fig. 14), suggesting that E2F1 upregulation in P3 tumor cells results from the immune pressure–driven enrichment of pre-existing E2F1+ tumor cells during immunotherapy. Collectively, our findings suggest that immune pressure induced by immunotherapy drives the selection of E2F1+ tumor cells, which subsequently upregulate HMGCR expression and contribute to ferroptosis resistance.

### The E2F1-HMGCR axis is associated with poor response to anti-PD-1 therapy

To determine the clinical relevance of the E2F1-HMGCR axis in the response to ICB therapy, we used the transcriptome data from melanoma patients classified as R or NR to anti-PD-1 therapy[44]. We found that expression levels of both HMGCR and E2F1 were significantly higher in NR compared to R (Fig. 5a, b). Notably, we found a positive correlation between HMGCR and E2F1 expression levels in the patients (Fig. 5c), indicating that the E2F1-HMGCR axis is conserved in patients. Importantly, patients with combined E2F1high and HMGCRhigh level was positively associated with resistance to anti-PD-1 therapy than those with E2F1low and HMGCRlow level (Fig. 5d). Moreover, patients whose tumors with combined E2F1high and HMGCRhigh showed decreased overall survival compared to patients whose tumors were E2F1low and HMGCRlow (Fig. 5e). Taken together, our result indicate that the E2F1-HMGCR axis could be a biomarker in predicting response and clinical outcome to anti-PD-1 therapy and has the potential as a target for therapeutic interventions aimed at restoring effective antitumor immunity and improving patient prognosis.

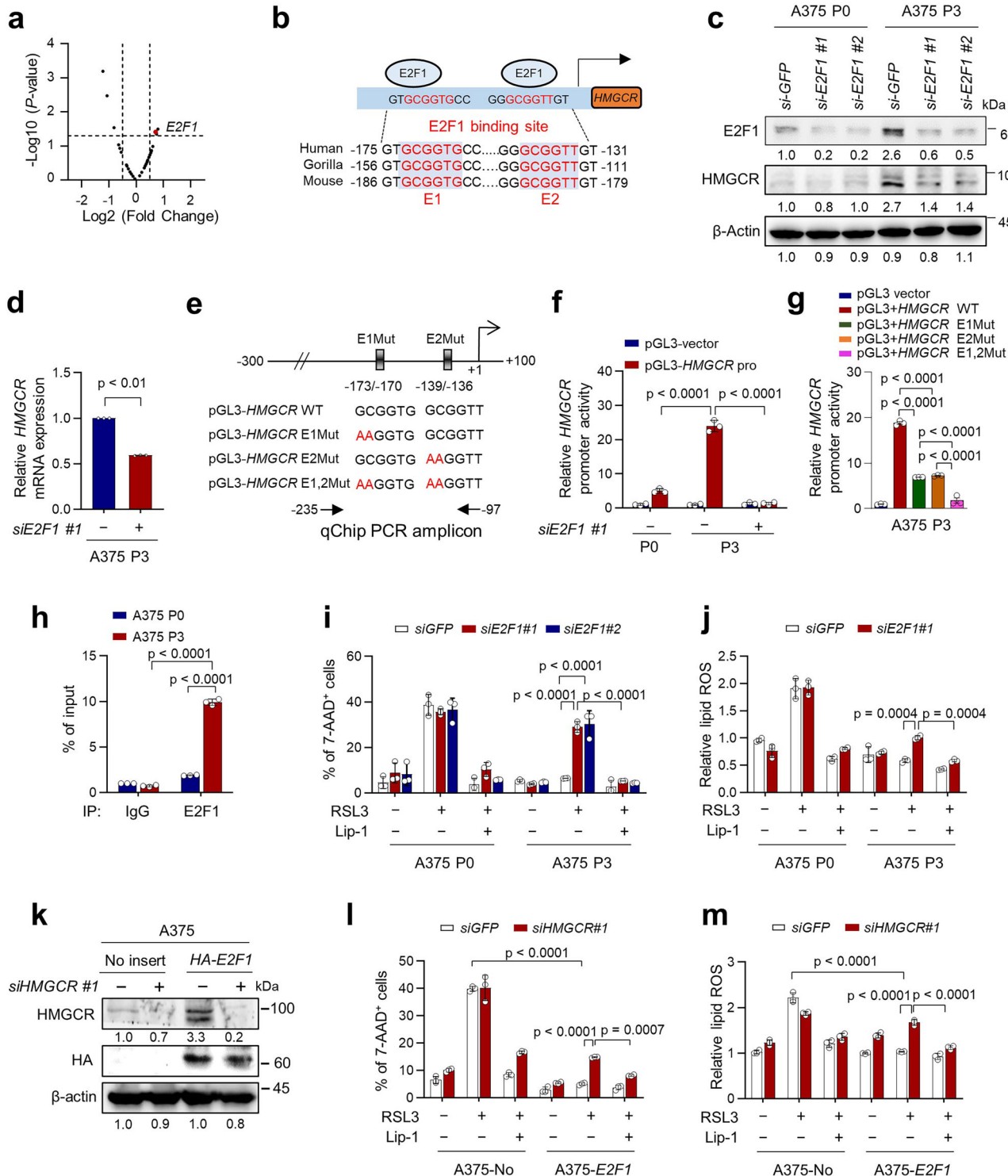

### Simvastatin sensitizes E2F1⁺ immune-refractory tumor cells to ferroptosis by inhibiting the HMGCR–CoQ10 axis

As E2F1-HMGCR is critical for ferroptosis-resistance of immune-refractory tumor cells, we reasoned that pharmacological inhibition of HMGCR might be an effective strategy for targeting immune-refractory tumor cells that highly express E2F1. To evaluate this idea, we chose simvastatin, a clinical HMGCR inhibitor used to reduce the risk of cardiovascular disease[45]. Notably, simvastatin synergized with RSL-3 in immune-refractory P3 cells but not in parental P0 cells (Fig. 6a, b). Consistent with our findings from genetic inhibition of HMGCR, simvastatin treatment enhanced ferroptosis in

immune-refractory P3 cells induced by either RSL3 or CTL-derived supernatant, as evidenced by increased levels of 7-AAD⁺ tumor cells and lipid ROS (Fig. 6c, d and Supplementary Fig. 15). These effects were abolished by IFNγ-neutralizing antibodies or Lip-1, supporting a critical role for soluble IFNγ in CTL-induced ferroptosis. This aligns with previous findings that IFNγ from antigen-specific T cells promotes ferroptosis via STAT1-mediated downregulation of SLC7A11/SLC3A2 and enhanced lipid peroxidation[14]. However, as IFNγ blockade substantially reversed ferroptosis, a residual effect persisted, suggesting the involvement of other factors involved. Since we observed that E2F1 over-expression in A375 P0 cells phenocopied ferroptosis-resistance of

**Fig. 4 | E2F1 directly regulates HMGCR, thereby promoting ferroptosis resistance of immune refractory tumor cells. a** The expression of HMGCR upregulator in non-responders relative to responders to anti-PD-1 therapy. The horizontal dashed line indicates *p*-value cutoffs at the 0.05 level. Red dot indicates the E2F1 gene. Vertical dashed lines indicate a fold change cut off (1.5FC). The *p*-values were determined by unpaired, two-tailed Student's *t* test. **b** Schematic diagram of E2F1 binding site on *HMGCR* promoter and sequence homology across different species. **c** The E2F1 and HMGCR proteins in A375 P0 and P3 cells transfected with siRNA targeting *GFP* or *E2F1* were detected using Western blot. **d** The HMGCR mRNA levels in A375 P3 transfected with siRNA targeting *GFP* or *E2F1*. **e** Diagram of the human HMGCR promoter region containing the E2F1 binding elements. The arrows indicate the ChIP amplicons corresponding to the two adjacent E2F1-binding sites. **f** Luciferase activities in A375 P0 and P3 cells first transfected with siRNA targeting GFP or E2F1 then transfected with the pGL3-basic or pGL3-*HMGCR*

WT plasmid as indicated. **g** Luciferase activities in A375 P3 cells transfected with the pGL3-basic, pGL3-*HMGCR* WT, pGL3-*HMGCR* E1Mut or E2Mut plasmids. **h** ChIP assay was carried out using A375 P0 and P3 cells. Cross-linked chromatin was immunoprecipitated with an anti-E2F1 antibody or an IgG control. The value of ChIP data represents relative ratio to the input, followed by qPCR analysis. Flow cytometry analysis of the 7AAD⁺ cells (**i**) and relative lipid ROS (**j**) in cells treated for 24 h with RSL3 (200 nM) with or without Lip-1 (1 μM) for an additional 20 h. HMGCR and HA-E2F1 protein levels (**k**), analyzed by western blots, and the percentage of 7-AAD⁺ cells (**l**) and relative lipid ROS (**m**), measured by flow cytometry, in A375-no insert and A375-HA-E2F1 cells transfected with *siGFP* or *siHMGCR#1* as indicated. All in vitro experiments were performed in triplicate. The error bars represent mean ± SD. All *p* value are presented exactly in figure. The *p* values by unpaired, two-tailed Student's *t* test (**d, g**) and two-way ANOVA (**f, h, i, j, l, m**) are indicated. Source data are provided as a Source data file.

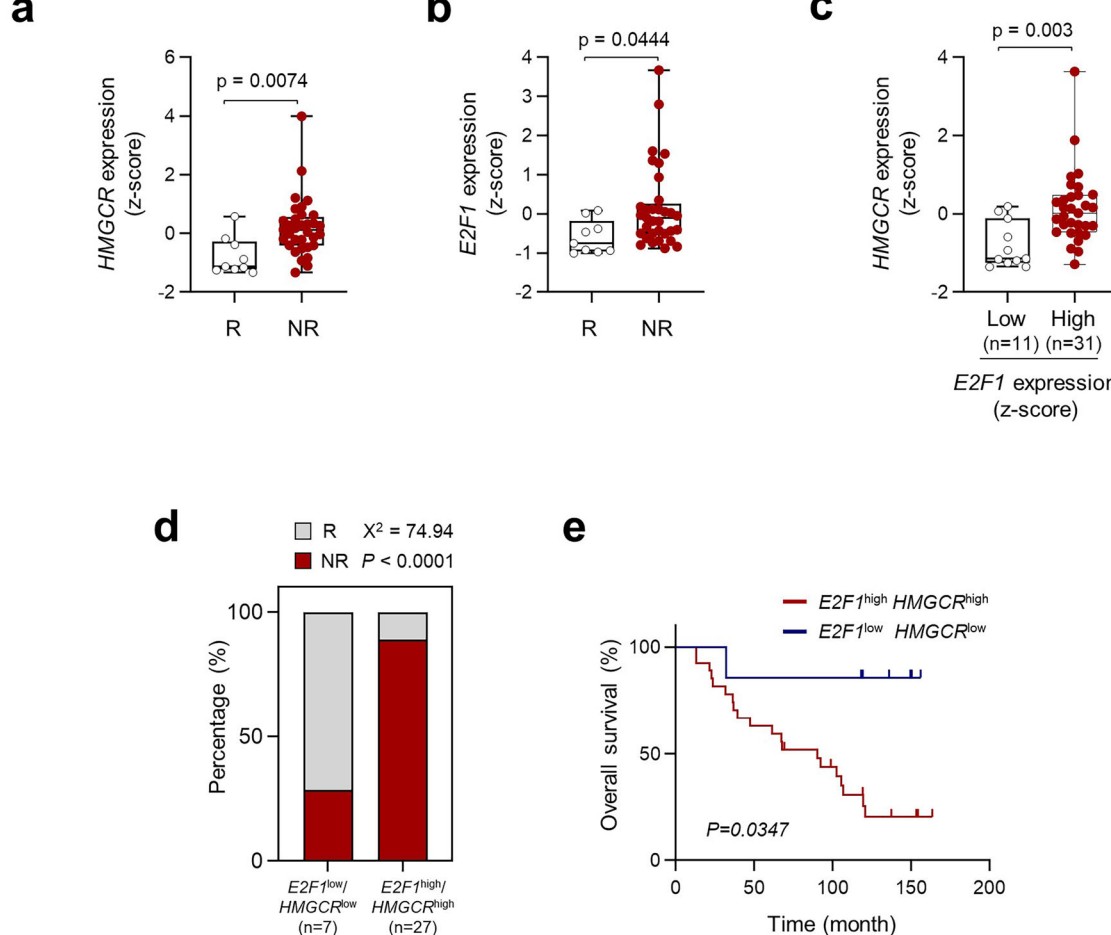

**Fig. 5 | E2F1-HMGCR axis is associated with poor response to anti-PD-1 therapy.** Comparisons of expressions levels of *HMGCR* (**a**) or *E2F1* (**b**) in responder (R, *n* = 9) and non-responder (NR, *n* = 33). **c** Comparisons of *HMGCR* expression in patients with low levels (*n* = 11) and high levels (*n* = 31) of *E2F1*. **d** Combined level of E2F1^high/ HMGCR^high was significantly associated with resistance to anti-PD-1 therapy in patients. **e** Kaplan–Meier analysis of overall survival (calculated as months to death or the last follow-up) of patients whose melanoma have low levels (*n* = 7) or high levels (*n* = 27) of both *E2F1* and *HMGCR*. In the dot plots, the center lines indicate the medians; and the ends of the whiskers indicate the maximum and minimum values,

respectively. The top and bottom edges of boxes indicate the first and third quartiles, respectively. **c–e** Optimal cut-off values were determined by performing receiver operating characteristic (ROC) analysis using MedCalc statistical software. The error bars represent mean ± SD. All *p* value are presented exactly in figure. **a–c** The *p* values by unpaired, two-tailed Student's *t* test are indicated. **d, e** The *p*-values were analyzed by a Student's two-side *t* test. The *p*-values were determined by Mann–Whitney *U* test (**d**) and *Gehan–Breslow–Wilcoxon test* (**e**). Source data are provided as a Source Data file.

A375 P3 cells (Fig. 4l, m), we then asked whether simvastatin could reverse the ferroptosis resistant properties of E2F1 overexpressed tumor cells. Indeed, simvastatin augmented RSL3-induced ferroptosis in E2F1-overexpressing A375 P0 cells but not A375 P0 cells (Supplementary Fig. 16). These results suggest that simvastatin is an effective

drug to reverse the ferroptosis resistance in E2F1^high immune refractory tumor cells.

We next investigated the downstream pathways of HMGCR that contribute to ferroptosis resistance. The mevalonate pathway is known to play a critical role in GPX4 maturation by supplying

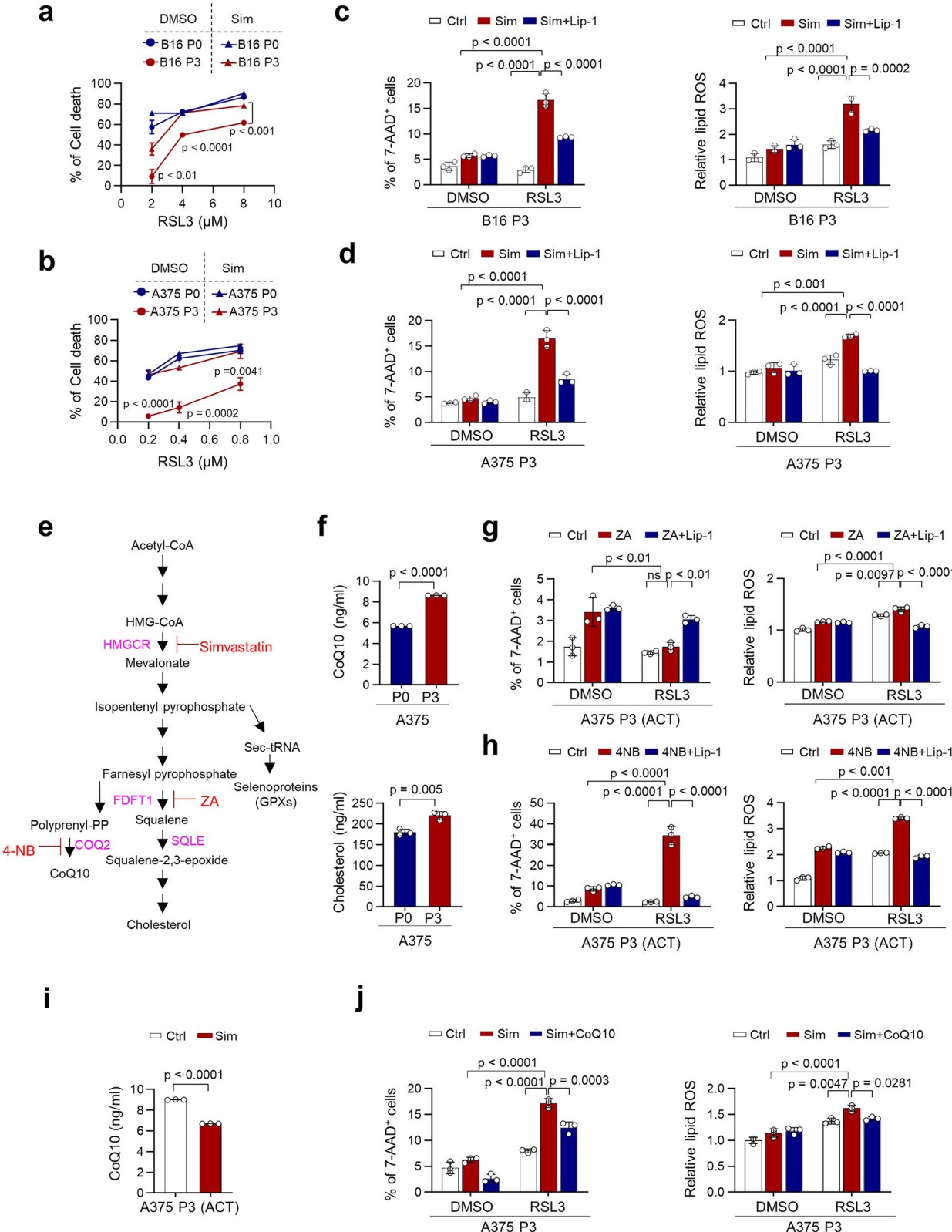

isopentenyl pyrophosphate (IPP)[46]. However, we observed no changes in GPX4 expression or enzymatic activity in P3 cells compared to P0 cells (Supplementary Figs. 3a–c, 8a, b), or following HMGCR inhibition by either simvastatin or siHMGCR (Supplementary Figs. 5, 17a, b), suggesting that HMGCR upregulation may confer ferroptosis resistance in immune-refractory tumors via GPX4-independent branches of the mevalonate pathway. Instead, we found that immune-refractory P3

tumor cells exhibited elevated levels of CoQ10 and cholesterol (Fig. 6e, f). Interestingly, the ferroptosis-refractory phenotype of A375 P3 cells was reversed by treatment with 4-nitrobenzoate (4-NB), a CoQ10 synthesis inhibitor, but not by zaragozic acid (ZA), an inhibitor of squalene synthase involved in cholesterol synthesis (Fig. 6g, h), indicating that CoQ10, rather than cholesterol, plays a critical role in HMGCR-mediated ferroptosis resistance. Notably, simvastatin

**Fig. 6 | Simvastatin overcomes ferroptosis resistance in Immune-refractory tumors via Inhibition of the HMGCR–CoQ10 Axis.** Cell death percentage of B16 P0 and P3 cells (**a**) or A375 P0 and P3 cells (**b**) treated with indicated concentrations of RSL3 with or without simvastatin (Sim, 1 μM) for 24 h. The percentage of cell death was determined by trypan blue exclusion assay. The percentage of 7AAD$^+$ cells or relative lipid ROS in B16 P3 cells (**c**) and in A375 P3 cells (**d**) treated for 48 h with RSL3 and Sim with or without Lip-1 (1 μM) for an additional 20 h. **e** The mevalonate pathway and its metabolic branches in mammalian cells, highlighting key enzymatic steps catalyzed by HMGCR, COQ2, FDFT1, and SQLE, along with their respective inhibitors. **f** The levels of CoQ10 and cholesterol in A375 P0 and P3 cells.

The percentage of 7AAD$^+$ tumor cells and lipid ROS levels were measured in A375 P3 cells treated with RSL3 (200 nM) alone or in combination with ZA (20 μM) (**g**) or 4-NB (1 mM) (**h**), with or without Lip-1 (1 μM). **i** The levels of CoQ10 in A375 cells treated with or without simvastatin. **j** The percentage of 7AAD$^+$ tumor cells and relative lipid ROS levels were assessed in A375 P3 cells treated with RSL3 (200 nM) alone or in combination with Sim (1 μM), with or without CoQ10 (200 μM) supplementation. All in vitro experiments were performed in triplicate. The error bars represent mean ± SD. All p value are presented exactly in figure. The *p* values by two-way ANOVA (**a**–**d**, **g**, **h**, **j**), and unpaired, two-tailed Student's *t* test (**f**, **i**) are indicated. Source data are provided as a Source data file.

treatment did reduce CoQ10 levels in P3 cells, and supplementation with CoQ10 reversed the enhanced ferroptosis sensitivity induced by simvastatin (Fig. 6i, j). These findings underscore the pivotal role of the HMGCR–CoQ10 axis in regulating ferroptosis resistance in immune-refractory tumor cells.

### HMGCR inhibition renders the immune-refractory tumor susceptible to T cell-based immunotherapy

Given our results in vitro, we reasoned that the in vivo administration of simvastatin could enhance therapeutic efficacy of T cell-based immunotherapy in immune-refractory tumor models by promoting tumor ferroptosis. To evaluate this possibility, CTL-refractory NYESO1$^+$ A375 P3 tumor-bearing NOD-SCID mice were administered the NYESO1-specific CD8$^+$ T cells alone or combined with simvastatin (Fig. 7a). While the T cell alone had no impact on tumor growth and simvastatin elicited a moderate therapeutic effect (Fig. 7b), combined therapy with CTLs and simvastatin profoundly retarded tumor growth, which was attenuated by Lip-1 treatment (Fig. 7b). Moreover, 7AAD$^+$ tumor cells and lipid ROS levels were significantly higher in the co-treated group than in other groups (Fig. 7c, d). We then examined the therapeutic potential of simvastatin in the treatment of ICB-refractory tumors by treating ICB-refractory B16 P3 tumor-bearing mice with PD-1 blockade alone, simvastatin alone, or both (Fig. 7e). Compared to treatment with anti-PD-1 or simvastatin alone, the combined therapy showed a remarkable therapeutic effect in B16 P3 tumor-bearing mice, which was attenuated by Lip-1 treatment (Fig. 7f). The combined treatment effect was accompanied by increased cell death and lipid ROS levels (Fig. 7g, h). To confirm the role of CD8$^+$ T cells in the observed therapeutic effect, we depleted them using an anti-CD8 antibody. This depletion markedly reduced the therapeutic benefits of the simvastatin and anti-PD-1 combination treatment (Supplementary Fig. 18), indicating that the enhanced immunotherapeutic effect of this combination strategy is largely dependent on CD8$^+$ T cells. Accumulating evidence suggests that tumor cell ferroptosis can promote antitumor immunity by releasing damage-associated molecular patterns (DAMPs)[47,48]. In our study, combined treatment with simvastatin and either RSL3 or T cell supernatant increased HMGB1 secretion and CRT exposure in B16 P3 and A375 P3 cells (Supplementary Fig. 19). These effects were abolished by co-treatment with Lip1, indicating that the DAMP release was dependent on ferroptosis. Consistent with our in vitro findings, simvastatin-treated mice showed significantly higher serum levels of HMGB1 and a greater proportion of CRT$^+$ tumor cells compared to controls (Fig. 7i, j), along with increased recruitment of conventional DCs to the tumor (Fig. 7k). These effects were abolished by co-treatment with the ferroptosis inhibitor Lip-1, confirming that DAMP release and DC recruitment were dependent on ferroptosis (Fig. 7i–k). Furthermore, the numbers of infiltrated functional CD8$^+$ T cells were higher in dual therapy groups than in monotherapy groups (Fig. 7l, m). These findings indicate that inhibiting HMGCR sensitizes immune-refractory tumor cells to ferroptosis-associated immunogenic cell death, thereby facilitating the activation of anti-tumor immunity. To evaluate potential safety concerns regarding the application of HMGCR inhibitors in treatment, we monitored body weight across all treatment groups and found no significant differences (Supplementary Fig. 20a). After two weeks of treatment,

simvastatin-treated mice showed a transient elevation in ALT and AST levels compared to the ICB monotherapy group, which returned to baseline within two weeks after treatment cessation (Supplementary Fig. 20b, c), consistent with previous clinical observations[49,50]. These results suggest that the simvastatin dose used in this study does not induce systemic or hepatic toxicity. Taken together, our data indicate that HMGCR inhibition with simvastatin can reverse ferroptosis resistance of immune-edited tumor cells, thereby enhancing the therapeutic efficacy of T-cell based immunotherapy, such as ACT and ICB therapy.

## Discussion

Cancer immunoediting is intricately associated with the emergence of tumor cells resistant to T cell-based therapies such as ACT and ICB[51]. The process of immune-selection fosters the development of tumor cells towards immune-refractory tumor cells with heightened resistance to CTL-mediated apoptosis, and cross-resistance to complement-dependent cytotoxicity[6,7,52]. Of the various cell death mechanisms aberrantly modulated by T cell-based therapy[53], ferroptosis had been implicated as an anti-tumor mechanism induced by CD8$^+$ T cells[14]. However, the details surrounding the modulation of ferroptosis in immune-refractory tumor cells has not been thoroughly explored. In this study, we show that CTL-mediated immune pressure confers resistance to ferroptosis in tumor cells through transcriptional induction of HMGCR by E2F1, thereby could contribute to refractory phenotypes to T cell-based therapies. While lipid peroxidation of immune-sensitive tumor cells is induced by supernatant derived from CTLs, it is not induced in immune-refractory tumor cells, indicating that tumor cells have evolved to evade ferroptosis under immunoediting. Although recent studies have identified several mechanisms that could protect tumor cells from unwanted ferroptosis, mechanisms responsible for linking immune editing to the acquisition of ferroptosis resistance remains to be elucidated. Here, we report that immune pressure imposed by T cell-based therapies drives the upregulation of HMGCR, a rate-limiting enzyme in the mevalonate pathway. Notably, after HMGCR depletion, ferroptosis-refractory properties of immune-edited P3 cells are almost completely lost, suggesting the crucial role of HMGCR in conferring ferroptosis resistance of tumor cells after immunoediting. Given the crucial role of HMGCR in the mevalonate pathway, an important unanswered question is how this pathway contributes to ferroptosis resistance in tumor cells during cancer immunoediting. While it is well established that the mevalonate pathway supports ferroptosis resistance by supplying isopentenyl pyrophosphate (IPP) for GPX4 production[46], our study found no changes in GPX4 levels or activity in P3 cells or following HMGCR inhibition. This suggests that alternative mechanisms may underlie ferroptosis resistance. Notably, the FSP1–CoQ10 pathway has been identified as an independent mechanism from the classical GSH–GPX4 axis for suppressing lipid peroxidation and ferroptosis[36]. In line with this, we observed elevated levels of CoQ10 and cholesterol in immune-refractory P3 tumor cells; however, only CoQ10 inhibition–not cholesterol inhibition–reversed ferroptosis resistance. These findings highlight CoQ10 as the key mediator of HMGCR-driven ferroptosis resistance. Thus, our data suggest that HMGCR may promote ferroptosis resistance in immune-edited tumor cells by enhancing CoQ10

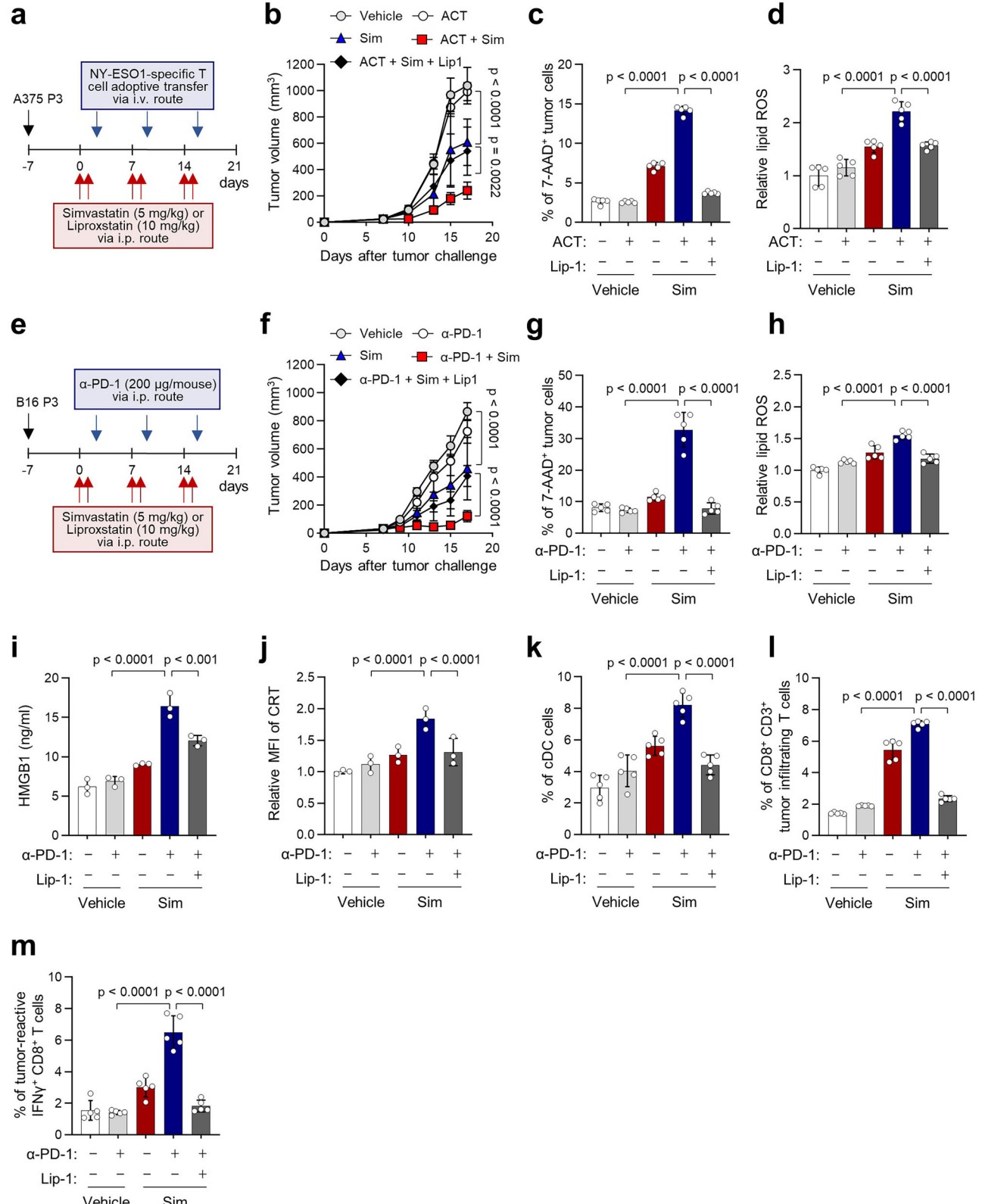

biosynthesis. Although FSP1 expression remained unchanged in our models, its role in reducing CoQ10 to ubiquinol suggests that targeting FSP1, in addition to HMGCR, may help overcome ferroptosis resistance in immune-refractory tumors.

Importantly, we notice that immune-edited P3 cells, in contrast to their parental (P0) cells, exhibit immune-refractory features of TME as well as a resistant phenotype to CTL-induced ferroptosis. Indeed,

knockdown of HMGCR significantly increases the overall number of tumor-infiltrating CD8+ T cells (TILs) or tumor-reactive CD8+ TILs in P3 tumors treated with anti-PD-1 Abs, which is accompanied by increased ferroptosis. However, HMGCR knockdown only retards tumor growth, without any ferroptosis features on the tumor cells, suggesting that the increased tumor reactive TILs is not from the direct anti-tumor effect by the silencing of HMGCR but rather from the increased susceptibility

**Fig. 7 | HMGCR inhibition renders the tumor susceptible to ferroptosis response to T cell mediated immune therapy. a** Schematic of the therapy regimen in NOD/SCID mice implanted with A375 P3 cells. Tumor growth (**b**) and the percentage of 7AAD$^+$ tumor cells (**c**) and lipid peroxidation (**d**), determined by flow cytometry, in mice inoculated with A375 P3 cells and treated with Sim or Lip-1 with or without NY-ESO1-specific T cell adoptive transfer (ACT). **e** Schematic of the therapy regimen in C57BL6 mice implanted with B16 P3 cells. Tumor growth (**f**), the percentage of 7AAD$^+$ tumor cells (**g**), lipid peroxidation (**h**), the level of HMGB1 (**i**), and CRT (**j**) determined by flow cytometry, flow cytometry profiles of tumor-infiltrating cDC cells (**k**), CD8$^+$ T cells (**l**), and the percentage of IFNγ$^+$ in CD8$^+$ T cells (**m**) in mice inoculated with B16 P3 cells and treated with the indicated reagents. For the in vivo experiments, 7 mice from each group were used, and randomly selected 5 samples were analyzed (**b**–**d**, **f**–**m**). All p value are presented exactly in figure. The p values by two-way ANOVA (**b**, **f**), and one-way ANOVA (**c**, **d**) and (**g**–**m**) are indicated. The error bars represent mean ± SD. Source data are provided as a Source data file.

of tumor cells to ferroptosis induced by tumor-reactive T cells primed upon PD-1 blockade. It is intriguing that gain of HMGCR appears to potentiate an immune-refractory TME during ICB therapy. Accumulating evidence indicates that intrinsic resistance to CTLs can be a crucial cause to disrupt a self-sustaining cancer immunity cycle, which determines immune phenotypes, called hot and cold tumors[54]. The involvement of the HMGCR-mediated mevalonate pathway in the anti-tumor immune responses has been implicated by previous research, which demonstrated increased immune responses to tumor cells undergoing vaccine adjuvants and cancer immunotherapies following interruption of the mevalonate pathway[55]. Thus, our present study demonstrates that HMGCR upregulation in tumor cells subjected to immune editing by ICB therapy promotes intrinsic resistance to ferroptosis, which in turn could contribute to immune-refractory TME.

The upregulation of HMGCR has been reported in multiple tumors[56,57]. However, the regulatory mechanisms of HMGCR expression, particularly in the course of CTL-mediated immune selection, has not yet been extensively studied. In this study, we report E2F1 as an upstream regulator of HMGCR, which is responsible for ferroptosis resistance of immune-edited tumor cells. Interestingly, the E2F1-binding element in the HMGCR promoter region is evolutionarily conserved, suggesting the important role of E2F1 in HMGCR regulation. Indeed, compared to their parental P0 cells, E2F1 is upregulated in immune-edited P3 cells, and knockdown of E2F1 in P3 cells reduces HMGCR levels and increases their susceptibility to ferroptosis. While we have established that the E2F1-HMGCR axis drives ferroptosis resistance, this raises an important question: how is E2F1 upregulated during cancer immunoediting? In this regard, our previous studies demonstrated that CTL-mediated immune selection enriches immune-refractory cancer cells with CSC-like properties and resistance to CTL-mediated apoptosis, driven by E2F1 through FGFR signaling or NANOG induction. NANOG further amplifies E2F1 activity via AKT-mediated RB suppression, forming a reciprocal loop[42,43]. In light of this, our current study suggests that immune pressure imposed by immunotherapy drives the selective enrichment of pre-existing E2F1$^+$ tumor cells within the parental population, which in turn upregulate HMGCR expression and promote ferroptosis resistance. Building upon previous findings, we further establish E2F1 as a key regulator of both apoptosis and ferroptosis resistance in the context of tumor cell death mediated by CTLs reinvigorated through anti-PD1 therapy.

Importantly, HMGCR expression in melanoma patients was positively correlated with E2F1 expression, suggesting the molecular axis in tumor cells that were observed in vitro hold true for cancer patients. Moreover, we find that the expression degree of the E2F1-HMGCR axis is correlated with the responsiveness to the anti-PD-1 therapy and worse survival in patients. For the clinical application of treatment strategies combined with immunotherapy, the presence of biomarkers that predict response and clinical benefit is essential for selecting patients who most likely would benefit from the treatment. Given an essential role of ferroptosis in the anti-tumor immunity induced by T cell-based therapies, our finding provides a strong rationale that the E2F1-HMGCR axis can be used to predict the clinical outcomes of ICB therapy.

Statins, HMGCR inhibitors, are currently used to treat dyslipidemia and are being explored for potential combining with anti-cancer drugs[45,55]. Notably, we demonstrate that simvastatin treatment

synergizes with T cell-based immunotherapy, including anti-PD-1 therapy and ACT, in reversing the resistance to ferroptosis in the immune-refractory tumors. Therefore, our results provide a mechanistic rationale for combining statins with T cell-based therapy to control immune-refractory tumors. The expression of PD-L1 in tumors and the presence of tumor-infiltrating lymphocytes (TILs) are well known as companion diagnostic biomarkers to prescribe ICB[58]. Given the evidence that inhibiting HMGCR with simvastatin remodels non-T cell-inflamed (TIL$^-$) tumors into T cell-inflamed (TIL$^+$) tumors, treatment of patients having PD-L1$^+$/TIL$^-$ or PD-L1$^+$/HMGCR$^+$ tumors with simvastatin may convert the tumors to TIL$^+$, providing clinical benefits to ICB therapy. Taken together, considering that the number of the FDA-approved statins including simvastatin, pravastatin, fluvastatin, lovastatin, atorvastatin, rosuvastatin, and pitavastatin, the clinical application of the actionable statins as ferroptosis-sensitizing drugs to treat E2F1$^{high}$ and HMGCR$^{high}$ immune-refractory tumor cells may be a promising strategy in combating resistance to T cell-mediated immunotherapy and reinvigorating the cancer-immunity cycle for better clinical outcome.

It is intriguing, and perhaps counterintuitive, that HMGCR inhibition alone did not strongly induce classical features of ferroptosis, such as increased 7-AAD$^+$ cells and lipid ROS. However, when combined with PD-1 blockade, it did trigger ferroptosis in immune-refractory tumor cells, suggesting that HMGCR inhibition primes tumor cells for ferroptosis, which is subsequently activated during T cell-mediated immune attack facilitated by PD-1 blockade. Nonetheless, we do not exclude the possibility that ferroptosis-independent mechanisms also contribute to the antitumor effects of HMGCR inhibition. For instance, disrupting protein prenylation, a downstream process of the mevalonate pathway, can impair oncogenic signaling, reduce tumor cell proliferation, and enhance immunogenicity by interfering with RAS or RAC1 function[59–61]. Thus, the therapeutic benefits of HMGCR inhibition likely result from both ferroptosis sensitization under immune pressure and additional anti-proliferative or cytotoxic mechanisms, such as impaired prenylation, underscoring its multifaceted potential to enhance the efficacy of T cell-based immunotherapies.

It is important to note that the role of ferroptosis in promoting anti-tumor immune responses remains complex and controversial. While ferroptotic cancer cell death can trigger the release of damage-associated molecular patterns (DAMPs)—a hallmark of immunogenic cell death (ICD)[62]—and has been proposed as a strategy to enhance cancer immunotherapy[47,48], its immunogenic potential has shown variability across experimental models[63]. Moreover, recent studies underscore the dual role of ferroptosis in the tumor microenvironment: although ferroptosis in tumor cells may stimulate anti-tumor immunity, ferroptosis in immune cells such as CD8$^+$ T cells, NK cells, and neutrophils can suppress it[64–66]. In this context, our findings demonstrate that targeting HMGCR in immune-refractory tumor cells induces ferroptosis-associated ICD and enhances anti-tumor immune activation during T-cell based immunotherapy. While we cannot completely exclude the possibility that HMGCR inhibition may influence other immune cell populations, the observed increase in CD8$^+$ T cell activity suggests that simvastatin primarily promotes tumor cell-specific ferroptotic death without compromising CD8$^+$ T cell function. Nonetheless, the successful clinical translation of HMGCR

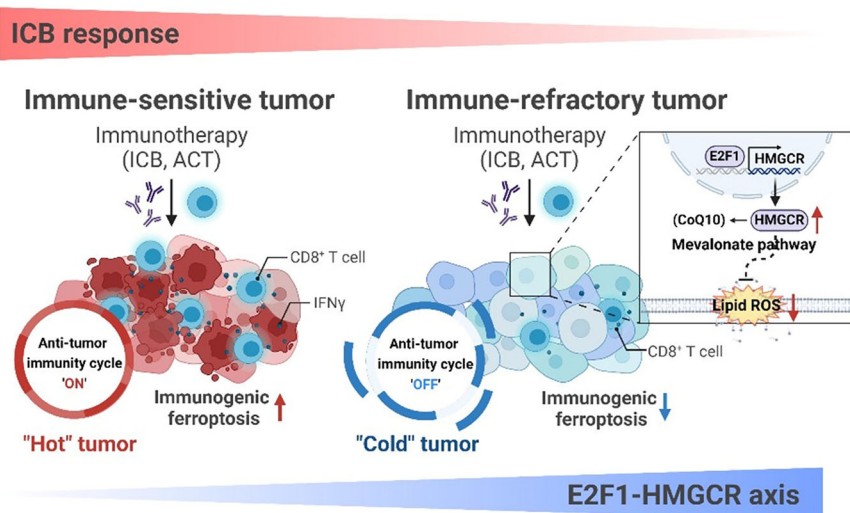

**Fig. 8 | Scheme for the role of E2F1-HMGCR axis in ferroptosis resistance in immune refractory tumor cells.** E2F1-HMGCR axis drives refractoriness against immunotherapy by triggering ferroptosis resistance.

inhibitors as immunotherapy adjuvants will require a deeper mechanistic understanding of how to selectively harness ferroptosis to stimulate anti-tumor immunity while minimizing unintended effects on immune effector cells.

During T cell-based cancer immunotherapy, ferroptosis of tumor cells by activated T cells can contribute to a sustained antitumor effect by activating anti-tumor immunity. However, tumor cells over the course of immunoediting may evade from CTL-mediated killing by deregulating intrinsic cell death mechanisms including ferroptosis, thereby disrupting the anti-tumor immunity cycle, which provokes resistance to the immunotherapy (Fig. 8). In this study, we propose the mechanism by which immune editing could promote resistance of ferroptosis in tumor cells. In this process, E2F1 upregulates HMGCR through promoter binding, thereby diminishing the oxidative damage induced by immunotherapy (Fig. 8). Furthermore, we demonstrate that inhibition of HMGCR with simvastatin enhances tumor susceptibility to T-cell-based immunotherapy by reversing ferroptosis resistance. Therefore, our data provide evidence that the inhibition of HMGCR may be a promising strategy that will help combat E2F1[high] immune-refractory tumors, particularly in regard to immune-based cancer therapy.

Study limitations in our work include the small number of clinical samples, which restricted our ability to fully investigate the relationship among the E2F1–HMGCR axis, ferroptosis resistance, and ICB response in patient tumors. Our study provides mechanistic insight into how T cell-based immunotherapies, such as PD-1 blockade and adoptive T cell transfer, promote ferroptosis resistance through immunoediting in tumor cells. By identifying the E2F1–HMGCR axis as a key driver of this resistance, we demonstrate that HMGCR inhibition restores ferroptosis sensitivity, rendering immune-refractory tumor cells susceptible to ACT and PD-1 blockade.

However, due to the limited clinical samples, we were unable to directly assess ferroptosis-related features, such as lipid peroxidation, in patient specimens. Future studies involving larger patient cohorts and immunohistochemical analysis of ferroptosis markers will be essential to validate the clinical relevance of our findings. In addition, single-cell transcriptomic profiling of immune-refractory tumor models and patient samples will provide high-resolution insights into cell-type-specific HMGCR functions and regulatory networks. This approach is expected to refine our understanding of ferroptosis resistance and reveal complex intercellular interactions within immune-refractory tumors, ultimately informing precision

immunotherapeutic strategies. Nonetheless, our findings provide a strong preclinical foundation for targeting ferroptosis resistance to improve the efficacy of T cell-based cancer immunotherapies.

## Methods

### Mice
Six- to eight-week-old female NOD/SCID or C57BL/6 mice were purchased from Central Lab. Animal Inc. (Seoul, Korea). All mice were maintained and handled under the protocol approved by the Korea University Institutional Animal Care and Use Committee (KUIACUC-2021-0049). All animal procedures were performed in accordance with recommendations for the proper use and care of laboratory animals.

### Cell lines
B16, TC-1, A375 cells were purchased from the American Type Culture Collection (ATCC, Manassas, VA, USA) between 2010 and 2014 and were tested for mycoplasma using a Mycoplasma Detection Kit (Thermo Fisher Scientific, San Jose, CA, USA). The identities of the cell lines were confirmed by short tandem repeat profiling by IDEXX Laboratories, Inc., and used within 6 months for testing. The generation of the immuno-refractory A375 P3 and B16 P3 cell lines was as described previously[31,40]. To generate ICB therapy–refractory tumor cells, TC-1, between 5 and 7 days of the tumor challenge, mice were treated with anti-PD-1 antibody (200 µg; BioXcell) three times per week. This treatment regimen was repeated for 2 cycles. The surviving P0 tumor was expanded in vitro. This escape variant cell line was designated TC-1 P1 and injected into a new group of mice and selected by anti-PD-1 therapy again. This process was repeated for 3 cycles to derive the P3 line, which was impervious to therapeutic effect by anti-PD-1. All cells were grown at 37 °C in a 5% $CO_2$ incubator/humidified chamber.

### Reagents
Erastin, RSL3, liproxstatin-1 and zaragozic acid A (ZA) were purchased from Cayman Chemical. 4-nitrobenzoate (4-NB) was purchased from Alfa Aesar. Simvastatin and CoQ10 were purchased from Sigma-Aldrich. Liperfluo was purchased from Dojindo Molecular Technologies.

### siRNA constructs
Synthetic small interfering RNAs (siRNAs) specific for *GFP*, *HMGCR*, *E2F1*, *Hmgcr* and *E2f1* were purchased from Bioneer (Korea): nonspecific *GFP* (green fluorescent protein), 5′-GCAUCAAGGUGAACUUCAA-3′ (sense), 5′-UUGAAGUUCACCUUGAUGC-3′ (antisense); *HMGCR*#1,

5′-CUGUCAAGUACAAGCAAGA-3′ (sense), 5′-UCUUGCUUGUACUUGA-CAG-3′(antisense); *HMGCR*#2, 5′-CGUCCAAUUUGGCAGCUCA (sense), 5′-UGAGCUGCCAAAUUGGACG-3′ (antisense); *E2F1*#1, 5′-ACGCUAUGA-GACCUCACUG-3′ (sense), 5′-CAGUGAGGUCUCAUAGCGU-3′ (antisense); *E2F1*#2,; *Hmgcr*#1, 5′-GACGAAGAAAACAGUCACU-3′ (sense), 5′-AGUGA-CUGUUUUCUUCGUC-3′ (antisense); *Hmgcr*#2, 5′-CGUCCAAUUUGG-CAGCUCA-3′ (sense), 5′-UGAGCUGCCAAAUUGGACG-3′ (antisense); *E2f1*#1, 5′-GUGGAUUCUUCAGAGACAU-3′ (sense), 5′-AUGUCUCUGAA-GAAUCCAC-3′ (antisense); *E2f1*#2, 5′-CCUGCAGAACAGAUGGUCA-3′ (sense), 5′-UGACCAUCUGUUCUGCAGG-3′ (antisense).

## DNA constructs and site-directed mutagenesis

The pCMV-*HA-E2F1* plasmid has been described previously[67]. The pCMV-*FLAG-HMGCR* plasmid was purchased from Sino Biological Inc. The promoter region of the human *HMGCR* gene was isolated with a PCR based strategy from genomic DNA extracted from A375 cells using a primer set, 5′-AACTGGTACCTGGGACTCGAACGGCTATTG-3′(forward) and 5′-GGCAAAGCTTACAGAATCCTTGGATCCTCC (reverse). The PCR products were digested with *Kpn*I and *Hind*III and subcloned into the *Kpn*I/*Hind*III restriction sites of the pGL3-Basic vector (Promega). To generate mutations in the E2F1 binding site of *HMGCR* promoter region, site-directed mutagenesis was performed using a QuickChange XL Site-directed Mutagenesis kit (Stratagene, San Diego, CA, USA) according to the manufacturer's instructions. The following primer sets were used: for pGL3-*HMGCR* promoter-E1 MUT, 5′- AGAGATGGTAAGGTGC CTGTTCTTGGCCC-3′ (sense) and 5′-ACAGGCACCTTACCATCTCTCAC-CACGGC-3′ (antisense). for pGL3-HMGCR promoter-E2 MUT, 5′- GTGGAAGGTTGTTAAGGCGACCGTTC-3′ (sense) and 5′- TTAACAAC CTTCCACAGCTCTCTGCAGG-3′ (antisense). Mutations were confirmed through DNA sequencing.

## Real-time quantitative RT-PCR

Total RNA was isolated using RNeasy Micro kit (Qiagen), and the cDNAs were synthesized by reverse transcriptase (RT) using Prime-Script RT Master Mix (Takara Bio Inc.), according to the manufacture' recommended protocol. Real-time quantitative PCR was performed using iQ SYBR Green super mix (Bio-Rad) with the specific primers on a CFX96 real-time PCR detection system. All real-time quantitative PCR experiments were performed in triplicate and quantification cycle (Cq) values were determined using Bio-Rad CFX96 Manager 3.0 software. Relative quantifications of the mRNA levels were performed using the comparative Ct method with β-ACTIN as the reference gene. Supplementary Table 1 describes the primers used for real-time PCR. Additional primer sets for genes responsible for negative regulation of ferroptosis were described in Supplementary Table 1.

## Western blot analysis

Lysate extracted from a total of $2 \times 10^5$ cells was used to perform Western blot. Primary antibodies against HMGCR (sc-271595, Santa Cruz Biotechnology, C-1), HA (sc-805, Santa Cruz Biotechnology, Y-11), FLAG (M185-7, MBL, FLA-1), E2F1 (sc-251, Santa Cruz Biotechnology, KH95), E7 (sc-6981, Santa Cruz Biotechnology, ED17), GPX4 (52455T, Cell Signaling Technology, polyclonal), SLC7A11 (12691T, Cell Signaling Technology, D2M7A), and β-ACTIN (sc-47778, Santa Cruz Biotechnology, C4) were used. Western blotting was followed by incubation with the appropriate secondary antibodies conjugated to horseradish peroxidase, anti-rabbit IgG-HRP (sc-2357, Santa Cruz Biotechnology, unknown clone), and anti-mouse IgG-HRP (Cat ADI-SAB-100-J, Enzo, polyclonal). For Western blotting, primary antibodies were used at a 1:1000 dilution and secondary antibodies at a 1:5000 dilution. Lot numbers were not specifically recorded during the experiments, but all antibodies were commercially sourced and consistently used throughout the study. Immunoreactive bands were developed with the chemiluminescence ECL detection system (Thermo scientific, USA),

and signals were detected using a luminescent image analyzer (SC-600MF, Davinch-K, Korea).

## Trypan blue exclusion assay

Aliquots of the B16, A375 and TC-1 cell suspensions were mixed with trypan blue dye and were left for 5 min at room temperature. Then, the B16, A375 and TC-1 cells were counted using a hemocytometer and the percentage of live cells was determined. Data are expressed as percentages of unstained cells compared with control cells not exposed to the chemical reagents.

## Glutathione assay

GSH/GSSG ratio was quantified according to the manufacturer's instructions using the in vitro GSH/GSSG (EGTT-100). To measure the concentration of GSSG were sonicated in 200 μl of cold buffer (50 mM phosphate (pH 7.0), 1 mM EDTA and 20 μl scavenger). The cell lysates were centrifuged at $10,000 \times g$ for 5 min at 4 °C and the supernatants were transferred to clean tubes for the deproteinization. For the determination of total glutathione (GSH) were sonicated in 200 μl of cold buffer (50 mM phosphate pH 7.0 and 1 mM EDTA). The samples were centrifuged at $10,000 \times g$ for 15 min at 4 °C. The supernatants were transferred to clean tubes to carry out the deproteinization using a solution of 5% metaphosphoric acid (MPA). A standard curve for GSH was prepared following the manufacturer's instructions and the absorbance was read at 412 nm at 0 min and 10 min. Assays were performed in triplicate. The results were expressed in μM of GSH/mg protein.

## In vitro cell death measurement

For cell death analysis, cells were treated with indicated reagents, collected, then resuspended in PBS containing 1 μg/ml 7-aminoactinomycin D (7-AAD), was obtained from Beckman Coulter. for 5 min, and directly run on a flow cytometer (Beckman coulter).

## In vitro immunogenic cell death measurement

For analysis of surface exposed CRT and HMGB1, $2 \times 10^5$ B16 P3 and A375 P3 cells treated for 48 h with RSL3 and simvastatin (1 μM) with or without liproxstatin (1 μM) for an additional 20 h. The cells were dissociated with accutase (00–4555-56, Thermo Fisher Scientific). CRT surface expression was estimated by flow cytometry using an anti-CRT antibody (Ab2907, Abcam, polyclonal) and a secondary antibody conjugated with Alexa-Fluor 488 (Molecular Probes-Invitrogen). For the study of HMGB1, samples from cell culture supernatant were concentrated by protein concentrator spin columns (UFC5003, Merck Milipore) before assay. We measured the concentration of HMGB1 using Human HMGB1 enzyme-linked immunosorbent assay (ELISA) kit (ARG81185, Arigo Biolaboratories Corp., Burlington, NC, USA) according to the manufacturer's instructions.

## Lipid peroxidation assessed by Liperfluo staining

For Liperfluo staining, cells were treated with RSL3 in the absence or presence of liproxstatin-1 for an additional 20 h, Cells were stained in Hanks balanced salt solution (HBSS, Gibco) containing Liperfluo (10 μM) for 30 min at 37 °C, collected by trypsinization, and analyzed immediately with a flow cytometer (Beckman coulter). In some cases, cells were pre-loaded with 10 μM Liperfluo for 1 h at 37 °C, followed with the indicated treatments. The fluorescence intensity in FITC channel was monitored. All data are shown as the relative mean fluorescence intensity (MFI) of FITC. The data were normalized to control samples as shown by relative lipid ROS.

## HMGCR CoA reductase activity assay

The activity of HMG-CoA reductase was measured basing on the consumption of NADPH by the enzyme, which can be measured by the decrease of absorbance at OD = 340 nm using the HMG-CoA

Reductase Activity Colorimetric Assay Kit, (ab204701; Abcam, Cambridge, MA) in accordance with manufacturer's directions. The total protein of cells was used for normalization.

## Luciferase assay

To determine the promoter activity of *HMGCR*, dual-luciferase reporter assay kits were used (Promega, E1910). Briefly, the reporter plasmid, pGL3 basic, pGL3-*HMGCR* WT, or pGL3-*HMGCR* MUT together with Renilla luciferase control plasmid, pGL4.74 [hRluc/TK] vector, an internal control for transfection efficiency, were co-transfected into A375 cells using Lipofectamine 2000 (Invitrogen). After 24 h, cells were washed with PBS and lysed with passive lysis buffer (Promega). Firefly luciferase activity was measured with a SpectraMax iD5 multimode microplate reader (Molecular Devices, San Jose, CA) after addition of 100 μl of luciferase assay reagent (Promega). Then, renilla luciferase activity was measured after addition of 100 μl of Stop & Glo reagent (Promega). The relative firefly luciferase activity in the reporter plasmid was normalized to that of Renilla luciferase.

## Chip and quantitative ChIP (qChIP) assays

The Chip assay in A375 cell lysates were measured using the ChIP kit (Thermo Scientific, Inc.) according to the manufacturer's instructions. Briefly, cells ($4 \times 10^6$ per assay) were incubated in 1% formaldehyde at room temperature for 10 min for cross-linking of proteins and DNA and then lysed in MNase digestion buffer containing Micrococcal nuclease. DNA was sheared by sonication using a Sonic Dismembrator Model 500 (Fisher Scientific, Pittsburgh, PA, USA). Immunoprecipitation was carried out by incubation with 5 μg of E2F1 (sc-251, Santa Cruz Biotechnology), or mouse IgG (Thermo Scientific, Inc.) for 16 h. For qChIP assay, immunoprecipitated DNA was quantified by real-time qPCR using following primer sets; 5'-GGCAGGCCCTAGTGCTGGGA-3' (forward) and 5'-GCCTGACGGCGCTACGTCACGAAC-3'(reverse). Each sample was assayed in triplicate, and the amount of precipitated DNA was calculated as the percentage of input sample.

## In vitro CTL mediated cell death assays

The A375 cells were incubated for 24 h with supernatant derived from NY-ESO1-specific CD8+ T cell lines. The frequency of ferroptotic cells was analyzed by staining with 7-aminoactinomycin D (7-AAD) or Liperfluo, and examined by flow cytometry. All analysis was performed using a Becton Dickinson FACSverse (BD Bioscience, USA).

## CTL-mediated apoptosis assay

TC-1 P0 and P3 cells were harvested by trypsinization and washed once with RPMI (Thermo Fisher, USA) containing 0.1% fetal bovine (FBS), resuspended, and labeled in 1 ml 0.1% FBS containing RPMI and 10 μM CFSE (Molecular Probes, Eugene, OR). Then, the CFSE-labeled cells were resuspended in 10 μg/ml E7 peptide containing 1 ml RPMI. After 1 h, CFSE-labeled TC-1 P0 and P3 cells were incubated for 4 h with an E7-specific CD8+T cell line at an E/T ratio of 1:1. After incubation for 4 h at 37 °C, the frequency of apoptotic cells was determined by staining with anti-active caspase-3 antibody (BD Pharmingen, Cat 561011) and performing flow cytometry. All analysis was performed using a Becton Dickinson FACSverse (BD Bioscience, USA).

## Immune profiling of tumor by FACS

The treated mice were euthanized on day 18 following tumor inoculation, and tumors were harvested. The tumors were dissected into fragments by cutting and digested by collagenase (0.5 mg/ml, MilliporeSigma) and DNase (1 μg/ml, MilliporeSigma) at 37 °C for 45 min. The digested samples were then filtered through a 100 μm cell strainer and washed with PBS buffer. The cell pellets were incubated with RBC lysis buffer to lyse the RBCs. The cell suspensions were stained for the intracellular and extracellular protein markers of interest, and the stained samples were assessed on a flow cytometer (BD biosciences) along with CellQuest Pro software. The following staining antibodies used: anti-CD3, anti-CD8 and anti-IFNγ (all from BD Biosciences).

## In vivo tumor treatment experiments

To characterize the in vivo resistance to CTL killing conferred by HMGCR, NOD/SCID mice were inoculated subcutaneously with $1 \times 10^6$ A375 P3 cells per mouse. Seven days following tumor challenge, simvastatin (5 mg/kg) or liproxstatin (10 mg/kg) was administered via Intraperitoneal injection for a day before adoptive transfer with NY-ESO1-specific CTLs (with 5000 unit IL-2) (Fig. 7a). This treatment protocol was repeated for 3 cycles. C57BL/6 mice were inoculated subcutaneously with $1 \times 10^6$ B16 P3 cells per mouse. Seven days following tumor challenge, simvastatin (5 mg/kg) or liproxstatin (10 mg/kg) was treated via intraperitoneal route every 3 days. Following day after simvastatin treatment, mice administered anti-PD-1 (BioXcell, NH, USA) or isotype antibody control that was administrated every 3 days at a dose of 200 μg per mice in accordance with the schedule described in Fig. 7e. In Fig. 2, C57BL/6 mice were inoculated subcutaneously with $1 \times 10^5$ B16 P3 cells per mouse. Seven days following the tumor challenge, *siGFP*- or *siHMGCR*#1, #2-loaded CNPs (5 μg/mouse) were administered via intravenous route (Fig. 2a and Supplementary Fig. 6). The day after siRNA treatment, the mice were administered 200 μg of anti-PD-1 (BioXcell, RMP1-14) or an isotype antibody control, which was administrated three times intraperitoneally every 3 days. This treatment regimen was repeated for three cycles. Mice were monitored for tumor burden for 20 days after challenge. Mice were handled and monitored for tumor burden and under the protocol permitted by the Korea University Institutional Animal Care and Use Committee (KUIACUC-2021-0049). Tumor size was measured before the tumors were smaller than, or at about 10% of mice body weight, the maximal tumor size/burden permitted by KUIACUC. In some cases, this limit has been reached on the last day of tumor size measurement and the mice were immediately euthanized.

To analyze immune and tumor cells within tumors, treated mice were euthanized and tumors were harvested and filtered through a 100 μm cell strainers and washed with phosphate-buffered saline (PBS) buffer. Cell pellets were then incubated with red blood cell (RBC) lysis buffer for 2 min. Cell suspensions were stained with specific protein markers, T cells including anti-CD45 (BD, #552848, K30-F11), anti-CD3 (BD, #555274, XMG1.2), anti-CD8 (BioLegend, #100731, 53-6.7), anti-IFN-γ (BD, #554412, XMG1.2), cDC including CD11c (TONBO, #35-0114, N418), CD103 (BioLegend, #121414, 2E7), MHCII (BD, #553570, AF6-88.5). For flow cytometry (FACS), antibodies were used at a 1:100 dilution. Lot numbers were not specifically recorded during the experiments, but all antibodies were commercially sourced and consistently used throughout the study. Stained samples were assessed on a flow cytometer Verse (BD Biosciences) and analyzed with Flow Jo (version 10.10.0). Flow cytometry gating strategies are included in Supplementary Fig. S21)

## Bioinformatic analyses from published clinical database

To investigate the clinical relevance in patients treated with anti-PD-1 therapy, we used the published datasets (GSE91061). Patients without cytolytic score, as a biomarker for antitumor immunity, are excluded from analysis. RNA-seq read counts were log2 normalized, and unbiased median gene expression value cutoffs were applied for the analysis of high/low expression subgroups and their potential associations with overall disease outcome (days to death or last follow-up). Transcription factors binding the promoter of HMGCR gene in low- or high-throughput transcription factor functional studies from the CHEA Transcription Factor Targets dataset (https://maayanlab.cloud/Harmonizome/gene/HMGCR).

## Statistics

All data are representative of at least three separate experiments. Statistical differences were calculated by either Student's *t* test (two-tailed, unpaired), one-way ANOVA, or two-way ANOVA using GraphPad Prism 10 (GraphPad Software, Inc). To divide into high- and low-risk groups, we performed receiver operating characteristic (ROC) analysis to identify optimal cut-off values using MedCalc statistical software (version 23.2.8). Survival curves were estimated using the Kaplan–Meier method, and differences between groups were assessed with the log-rank test. For all statistical tests, a *p*-value of less than 0.05 was considered indicative of statistical significance.

## Reporting summary

Further information on research design is available in the Nature Portfolio Reporting Summary linked to this article.

## Data availability

RNA sequencing data have been deposited in the Gene Expression Omnibus (GEO) under the accession code GSE91061. The CHEA Transcription Factor Binding Site analysis that support the findings of this study are publicly available online at [https://maayanlab.cloud/Harmonizome/gene/HMGCR]. Figures 1a–e, 1g–j, 2b–f, 3a–i, 4a, 4c, 4d, 4f–m, 5a–e, 6a–d, 6f–j, 7b–d and 7f–m, and Supplementary Figs. 1b–e, 2a, 2b, 3a, 3b, 4a, 4b, 5, 6b, 7a, 7b, 8a, 8b, 9a–c, 10a–d, 11a, 11b, 12a–c, 13, 14a–d, 15a, 15b, 16a–c, 17a, 17b, 18, 19a–c, 20a–c and 21 are provided in the Source Data file. The raw images for the immunoblots are provided in the supplementary information. All other data supporting the findings of this study are available from the corresponding author on reasonable request Source data are provided with this paper.

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

## Acknowledgements

This work was supported by funding from the National Research Foundation of Korea (RS-2021-NR061924, RS-2023-NR077043, NRF-2022R1A4A5032702, RS-2023-00280965, RS-2025-02306308, RS-2021-NR062038, RS-2022-NR067525). This manuscript was edited by Life Science Editors.

## Author contributions

Study concept and design: S.W.S., H.J.L, Y.-C.C., T.W.K., and K.H.S.; acquisition of data: S.W.S., H.J.L., N.K., S.B., E.C., H.K., D.Y.Y., C.L., J.C., S.L., and S.J.O.; analysis and interpretation of the data: S.R.W., Y.J.J., H.-J.C., J.Y.K., E.-W.L., J.P., S.K., M.K.S and S.G.K.; technical or other material support: Y.S., K.M.L., and C.Y.; writing and review of the manuscript: S.W.S., H.J.L., Y.-C.C., T.W.K., and K.H.S.

## Competing interests

The authors declare no competing interests.
