## [Transparent Peer Review file · Nature Communications]

The E2F1-HMGCR axis promotes ferroptosis resistance in immune refractory tumor cells

Corresponding Author: Professor Kwon-Ho Song

Version 0:

Reviewer comments:

Reviewer #1

(Remarks to the Author)

The study identifies the E2F1-HMGCR axis as a critical mechanism by which tumor cells acquire resistance to ferroptosis, thereby contributing to immune-refractoriness. This study also correlates findings in preclinical models with clinical data from melanoma patients, demonstrating the potential clinical significance of the E2F1-HMGCR axis in predicting and overcoming resistance to PD-1 blockade therapy. The research suggests that targeting HMGCR with inhibitors like simvastatin could enhance the efficacy of T cell-based immunotherapies. Overall, the study presents a significant advance in understanding ferroptosis resistance in immune-refractory tumors and offers a potential new avenue for improving cancer immunotherapy outcomes. However, additional research is needed to support their conclusions.

1. The study focused on ferroptosis and used Lip-1 as an inhibitor. However, early studies show that apoptosis resistance is a potential mechanism of this immune-refractoriness. It needs to use an apoptosis inhibitor in parallel as a control. Otherwise, it does not make sense to link or deny previous findings, namely that cytotoxic T cell responses promote apoptosis.
2. Although the study identifies E2F1 as a regulator of HMGCR, the upstream signals leading to E2F1 activation in the context of immune-refractory tumors are not fully elucidated.
3. The study demonstrates the efficacy of combining HMGCR inhibition with immunotherapy in mouse models. However, translating these findings to human patients requires careful consideration of potential side effects and the pharmacodynamics of HMGCR inhibitors in humans. Currently, there are no data provided by the authors on the toxicity effects in these animals.
4. The study focuses on the E2F1-HMGCR axis but does not extensively explore other potential mechanisms of ferroptosis resistance that might be relevant in different contexts or tumor types. The authors need to examine other ferroptosis resistance mechanisms, such as FSP1. It is also surprising that GPX4 is not changed in this study, as it is known that the mevalonate pathway can affect GPX4.
5. While the study shows increased infiltration of CD8+ T cells and enhanced anti-tumor immunity upon HMGCR inhibition, a more detailed analysis of the immune microenvironment (e.g., DAMP release, such as HMGB1) and other immune cell types (e.g., DC maturation and recruitment) could provide deeper insights into the overall immune response dynamics. Importantly, it needs to be confirmed whether the removal of CD8+ T cells will reverse the phenotype.
6. Non-cell-type selective induction of ferroptosis can dampen CTL activity. It remains unknown how targeting the HMGCR pathway only induces ferroptosis in cancer cells, rather than in CD8+ T cells.
7. All in vivo studies used only one siRNA; other siRNAs need to be repeated to confirm their findings.
8. The clinical data assay did not suggest a role for ferroptosis or lipid peroxidation. The number of patient samples used is low.

(Remarks on code availability)

Reviewer #2

(Remarks to the Author)

In this manuscript, Son et al. investigated whether and how ferroptosis plays a role in the anti-tumor activity of T cell-based immunotherapies by using immunotherapy-refractory tumor models. They showed that T cell-based immunotherapy drives the development of ferroptosis resistance of tumor cells. Furthermore, they showed that E2F1 was upregulated in immunotherapy-resistant cancer cells after immune editing, and it in turn bound to the promoter of the HMGCR gene to upregulate HMGCR, thereby contributing to the resistance to ferroptosis. Finally, they showed that HMGCR inhibition by siRNA or simvastatin rendered immune-refractory tumor cells susceptible to T cell therapy and PD-1 blockade therapy. Overall, this study provided a new mechanism by which cancer cells deregulate ferroptosis to escape immunotherapy-induced antitumor immunity.

Specific issues include:

1. Although ferroptosis is getting more attention in the cancer therapy field, T cell-induced cancer cell apoptosis are still considered the major mechanism of immune surveillance. Can the authors use some markers that specific for ferroptosis to demonstrate that the majority of cancer cells are undergoing ferroptosis or apoptosis during immune attack?
2. For Figure 1, did the authors detect differences in sensitivity to different death forms between P3 and P0 cell lines? Are there differences in other forms of cell death besides apoptosis and ferroptosis? Furthermore, did the authors assess the baseline expression levels of GPX4 and SLC7A11, which are critical factors controlling ferroptosis, between the P0 and P3 cell lines?
3. E2F1 has been involved in many important cellular functions including cell cycle progression and apoptosis. How could the authors exclude the possibility that E2F1 contributes to cancer cell resistance to immunotherapy by regulating cell cycle or apoptosis when under immune attack? Or by upregulating Nanog as in their previous publications (References 45,46)? Furthermore, it is still not answered that how E2F1 was upregulated from P0 to P3?
4. The downstream mechanism by which HMGCR downregulates ferroptosis sensitivity remain unclear. The authors only showed some preliminary data in Supplementary Figure S8 by using two chemical inhibitors. This is an important part of the study, and I would suggest put these results in a main figure, and rigorously perform CoQ10 and Cholesterol supplement and depletion experiment to verify their role in HMGCR-mediated ferroptosis resistance.
5. In Figure 2, the tumors in siHMGCR were significantly reduced compared with those in siGFP group. What was the underlying reason for tumor regression by HMGCR knockdown? Could lipro-1 rescue this part of the tumor regression?
6. HMGCR inhibition has been shown to increase cancer cell immunogenicity either by inhibiting RAS or RAC1 protein prenylation, and induce immunogenic cell death (e.g., PMID: 34330763, 34266895) . Did the authors observe changes of expression of certain immunogenic molecules such as HMGB1 and calreticulin after HMGCR inhibition on cancer cells? Whether blockade of ferroptosis would decrease the expression of these immunogenic molecules?
5. In Figure 3f, after E2F1 knockdown, the promoter activity of HMGCR was basically abolished, but why was the expression of Hmgcr in Figure 3g still much higher than that in the control group after mutation of the binding site?
6. The cells used in Figure 6b and Figure 6e were both A375 P0 cells, why were the sensitivity to statin killing obviously different?
7. The internal parameters (actin) depicted in Figure 1g exhibited inconsistencies, and the upregulation of HMGCR was not very pronounced. Similarly, the upregulation of HMGCR1 expression in Figure 3e was also not obvious. Quantification of the Western Blot (WB) results may enhance the credibility of these results.
8. Minor: There are a few writing mistakes throughout the manuscript, for example, the IC50 unit in Figure 1 should be "µm"; Please double check.

(Remarks on code availability)

Reviewer #3

(Remarks to the Author)

This is an interesting study shedding some light on the potential mechanism(s) of how statin drugs enhance responses to immunotherapy. The authors use B16 and TC-1 models, in addition to an adoptive cell transfer (ACT) mouse model, to show that immunotherapy resistance involves high expression of E2F1 and HMGCR leading to resistance to ferroptosis. Simvastatin or manipulation of E2F1/HMGCR are then able to restore ferroptosis, sensitivity to immunotherapy and CD8 T cell infiltration. Overall, this is a well performed study with appropriate controls, and it is overall well written.

Major concerns:

- 1) The Discussion is very limited. It includes a long summary of the results and potential implications, but there is little to no discussion of outside work and previously published studies for context. There are several preclinical and retrospective clinical studies showing that statins enhance responses to T-cell-based immunotherapy, and those should be cited here (and similarly the statement in the Introduction that "the impact of HMGCR on the resistance to anti-tumor immunity and ultimately to ICB is not yet known" is really not true). There is also no paragraph acknowledging limitations of the study.
- 2) More info is needed on the methods used to obtain data for Figure 5. What kind of cancers did these patients have? How was the 30% cutoff determined- was an ROC curve done? Does the Kaplan-Meier curve represent overall survival or progression-free survival?

Minor concerns:

- 1) The authors claim that ferroptosis can induce immunogenic cell death, but the literature on this is mixed. ICD is not just any type of cell death that promotes immunity, but a distinct process involving specific DAMPs (see studies by Lorenzo Galluzzi for example). Citing one paper and saying that ferroptosis induces ICD is not quite accurate.
- 2) Typo on page 14: "T cell-base" should be "T cell-based"

3) Are figures 6a, 6b and 6e also using trypan blue? This is worth mentioning again in the legend.

(Remarks on code availability)

Reviewer #4

(Remarks to the Author)

Son et al. studied the regulations of E2F1 and HMGCR on tumoral ferroptosis and immunotherapy resistance using both murine and human tumor and specific CTL models. They found that immunotherapy resistant tumor cells generated from in vivo ICB treatment became resistant to ferroptosis inducers in vitro. Mechanistically, these resistant cells express higher level of HMGCR and E2F1, and knockdown of these two genes could sensitize tumor to ferroptosis and ICB therapy. Meanwhile, they explored the regulatory mechanism underlying HMGCR transcription by E2F1. Simvastatin, as a HMGCR inhibitor, was shown to enhance tumoral ferroptosis and augment ACT- or ICB-mediated therapy. Overall, this study is well performed and the data is well organized. However, there are still several concerns need to be carefully addressed.

Major concerns:

1. The authors speculated that the CTL-resistant P3 cells became more resistance to ferroptosis, but how about apoptosis? The in vitro coculture of tumor and CD8 T cells is needed to test the difference between P0 and P3 cells against CTL cytotoxicity.
2. HMGCR is still expressed in P0 cells, but why its knockdown has no effect on RSL3-induced ferroptosis? Whether its over-expression affect ferroptosis?
3. The gating strategy of flow cytometry analysis for in vivo tumor cell death (Fig. 2c), lipid ROS (Fig. 2d) and T cell number (Fig. 2e) and cytokine production (Fig. 2f) are needed.
4. In the analysis shown in Fig.2e and 2f, why gp100 is used? In this case, the specific CTL recognize gp100 peptides should be detected.
5. What is the mechanism of E2F1 upregulation in P3 cells? Whether it is induced by in vitro CTL coculture or CTL supernatant?
6. The evidence showing the clinical relevance of the E2F1-HMGCR axis to ICB response is too weak. The IHC staining of HMGCR or E2F1 on tumor tissue slides is suggested. Or incorporate this part into Fig. 4 or Fig. 6.
7. Since simvastatin can promote ferroptosis, whether it can enhance tumor cell death in the co-culture with CTL?
8. Do the results shown in Supplementary Fig. 3a-b suggest that T-cell supernatant can directly induce ferroptosis of A375 P0 cells? How to explain this?

Minor points:

1. The cell death percentage shown in Fig. 4L is too low, suggesting no significant cell death was induced. This experiment needs to be repeated with higher RSL3 or longer treatment time.
2. The band of HMGCR shown in Supplementary Fig. 2a seems modified, please show the original exposure band of HMGCR.

(Remarks on code availability)

Reviewer #5

(Remarks to the Author)

(Remarks on code availability)

Version 1:

Reviewer comments:

Reviewer #1

(Remarks to the Author)

The authors have satisfactorily addressed my comments.

Reviewer #2

(Remarks to the Author)

The authors are appreciated for providing more experimental evidence that substantiate their hypothesis, and formally acknowledged and discussed the limitation of the current study. They have addressed most of my concerns. I still have two questions that need attention from the authors:

1. In response to my question 3 "it is still not answered that how E2F1 was upregulated from P0 to P3?". The authors proposed that immune killing pressure selected E2F1 high cancer cells. Do they think the E2F1 expression levels are heterogeneous among different cancer cell clones before immune selection, or E2F1 expression can be inducible during immune selection? Please be more specific.

2. If both apoptosis and ferroptosis contribute to immune killing of cancer cells, and E2F1 happens to regulate both cell death pathways, the authors need to strengthen the novelty of their findings in the context of cell death induced by T cells reinvigorated by anti-PD1 therapy.

Reviewer #3

(Remarks to the Author)

The authors have addressed all of my concerns, and the manuscript (especially the Discussion) is much improved.

Reviewer #4

(Remarks to the Author)

The authors have addressed my previous concerns by performing additional experiments or revising discussion. I have no more comments.

Dear Reviewers,

My coauthors and I appreciate the detailed and relevant comments that have been raised in response to our manuscript, "The E2F1-HMGCR axis promotes ferroptosis resistance in immune refractory tumor cells". Your assistance has been invaluable and has helped us improve the quality of our manuscript. We have fully addressed each reviewers' questions/comments and amended the manuscript content accordingly. We have also provided point-by-point responses to the critiques raised by the reviewers. All the reviewer questions are in ***bold italic text*** while our responses are in regular or underlined text (if necessary). In the revised manuscript, the changes in the revised manuscript are marked in RED. We hope this revision will clarify the points raised by the reviewers.

Once again, we thank you for your time, invaluable guidance, and consideration of our revised manuscript.

Reviewer Comments:

Reviewer #1 (Remarks to the Author): with expertise in ferroptosis, cancer

The study identifies the E2F1-HMGCR axis as a critical mechanism by which tumor cells acquire resistance to ferroptosis, thereby contributing to immune-refractoriness. This study also correlates findings in preclinical models with clinical data from melanoma patients, demonstrating the potential clinical significance of the E2F1-HMGCR axis in predicting and overcoming resistance to PD-1 blockade therapy. The research suggests that targeting HMGCR with inhibitors like simvastatin could enhance the efficacy of T cell-based immunotherapies. Overall, the study presents a significant advance in understanding ferroptosis resistance in immune-refractory tumors and offers a potential new avenue for improving cancer immunotherapy outcomes. However, additional research is needed to support their conclusions.

Major issues:

1. The study focused on ferroptosis and used Lip-1 as an inhibitor. However, early studies show that apoptosis resistance is a potential mechanism of this immune-refractoriness. It needs to use an apoptosis inhibitor in parallel as a control. Otherwise, it does not make sense to link or deny previous findings, namely that cytotoxic T cell responses promote apoptosis.

We thank the reviewer for the insightful comment. To investigate the role of ferroptosis and apoptosis in resistance to CTL-mediated cell death, we assessed both types of cell death in P0 and P3 cells by treating them with CTL-derived supernatant, which contains cytokines such as IFN γ previously shown to promote tumor cell death including ferroptosis and apoptosis (Wang, W. et al., Nature., 2019, Dragica et al., 2020, Biomarker Research, Hideo Komita., 2006, Journal of Hepatology), in the presence of the ferroptosis inhibitor Lip-1 and the apoptosis inhibitor zVAD. Compared to immune-susceptible P0 tumor cells, immune-refractory P3 cells exhibited resistance to both apoptosis and ferroptosis, as evidenced by reduced levels of active caspase-3⁺ cells and decreased lipid ROS production (revised Supplementary Fig. 2 and 7). Notably, in P0 cells, Lip-1 and zVAD partially suppressed CTL supernatant-induced ferroptosis and apoptosis, respectively, suggesting that both forms of cell death play roles in CTL-mediated tumor elimination. Furthermore, the Lip-1 almost completely reversed RSL3-induced cell death in P0 cells, whereas zVAD had no effect (revised Fig. 1c and 1d), demonstrating that RSL3 selectively induces ferroptosis. These results demonstrate that both apoptosis and ferroptosis contribute to CTL-mediated tumor cell death supporting our hypothesis that T cell-based immunotherapy-driven immune editing promotes tumor cell resistance to ferroptosis in response to cytotoxic T cell attacks. We have clarified these experimental rationales and findings in the revised Results sections for greater transparency.

[On Supplementary Fig. 2 and Supplementary Fig. 7 of the revised manuscript]

Supplementary Figure 2

Supplementary Figure 7

[On Fig. 1c, 1d of the revised manuscript]

Fig. 1

[On page 5, Results section of the revised manuscript]

Given ferroptosis, besides apoptosis, being a key anti-tumor mechanism promoted by CTL during ICB therapy¹⁴, we asked whether tumor cells acquire resistance to ferroptosis through ICB-mediated immune editing. In this regard, activated CTL-derived supernatants have been reported to promote both apoptosis and ferroptosis^{14,33,34}. To test this, we treated P0 and P3 cells with CTL-derived supernatants and assessed ferroptosis (characterized by lipid peroxidation) and apoptosis (marked by active caspase-3). Notably, CTL-derived supernatants increased the frequency of active caspase-3+ cells and elevated lipid ROS levels in immune-susceptible P0 cells (Supplementary Fig. 2a and 2b). These effects were partially blocked by the apoptosis inhibitor zVAD or the ferroptosis inhibitor Lip-1 (Supplementary Fig. 2a and 2b), indicating that both apoptosis and ferroptosis contribute to CTL-mediated tumor cell killing. In contrast, immune-refractory P3 cells exhibited resistance to both forms of cell death (Supplementary Fig. 2a and 2b). In addition, we compared the susceptibility of P3 versus P0 cells to RSL3 and erastin, known ferroptosis inducers that work through inhibiting GPX4 and cystine/glutamate antiporter (xCT), respectively^{19,35}. Based on the IC₅₀ values, indicating the concentration of drugs that inhibits 50% of cell viability, we found that P3 cells were less sensitive to the ferroptosis inducers than P0 cells (Fig. 1a and 1b). Indeed, RSL3-induced death in P0 cells was accompanied by increased lipid ROS production, both of which were reversed by Lip-1 but not by zVAD (Fig. 1c and 1d), indicating that RSL3 selectively induces ferroptosis in P0 cells. However, P3 cells did not show a significant difference in the cell death or the production of lipid ROS upon treatment with RSL3 (Fig. 1c and 1d), indicating ferroptosis resistance. These findings suggest that anti-PD-1 therapy-mediated immune selection promotes tumor cell resistance to ferroptosis in response to cytotoxic T cell attacks.

[On page 8, Results section of the revised manuscript]

We established an ACT-refractory A375 P3 model from the parental A375 cells (A375 P0) through three rounds of *in vivo* selection by HL-A2-restricted NYESO1-specific CD8⁺ T clone (MAK #11)⁴⁰. Intriguingly, like the ICB-refractory mouse tumor models, CTL-refractory human A375 P3 cells were more resistant to ferroptosis induced by either CTL-derived supernatant or RSL3 treatment than A375 P0 cells (Fig. 3a and Supplementary Figure 7a, 7b).

2. Although the study identifies E2F1 as a regulator of HMGCR, the upstream signals leading to E2F1 activation in the context of immune-refractory tumors are not fully elucidated.

The reviewer raised an important point regarding upregulation of E2F1 in immune-refractory tumors. In our previous study (Noh KH et al., J Clin Invest. 2012), we demonstrated that CTL-mediated immune selection promotes the emergence of cancer cells with enhanced survival advantages, leading to the enrichment of immune-refractory cancer cells characterized by unique features, including cancer stem cell (CSC)-like properties. In this regard, E2F1 contributes to CSC-like properties by activating FGFR signaling or inducing the stemness factor NANOG, and is further amplified through a reciprocal

regulatory loop in which NANOG-driven AKT hyperactivation impairs RB function, thereby enhancing E2F1 activity (Song KH et al., Oncogenesis 2017, Oh SJ et al., Cancer Res. 2018). Based on this, we hypothesized that the upregulation of E2F1 could be a consequence of immune selection pressure induced by ICB or ACT therapies. To investigate this, we evaluated E2F1 expression in TC-1 and A375 cells across successive rounds of immune selection (P0 to P3) and observed a stepwise increase in E2F1 levels from P0 to P3 cells (Revised Supplementary Fig. 14). These findings suggest that the elevated E2F1 levels in P3 cells are likely the result of enrichment of E2F1⁺ cells during the immune selection process. However, we do not exclude the possibility that both immune-driven enrichment of E2F1⁺ cells and heightened intrinsic E2F1 activity contribute to the elevated E2F1 levels and function seen in P3 cells. These mechanistic insights have been incorporated into the revised Results and Discussion sections to improve clarity and completeness.

[On Supplementary Fig. 14 of the revised manuscript]

Supplementary Figure 14. Son et al

[On page 10, Results section of the revised manuscript]

We next investigated the underlying mechanism responsible for the upregulation of E2F1 during immune editing. Our previous study showed that CTL-mediated immune selection enriches immune-refractory cancer cells with cancer stem cell (CSC)-like properties^{9,42}. In this regard, E2F1 contributes to CSC-like properties by activating FGFR signaling or inducing the stemness factor NANOG, and is further amplified through a reciprocal regulatory loop in which NANOG-driven AKT hyperactivation impairs RB function, thereby enhancing E2F1 activity^{42,43}. Based on this, we hypothesized that E2F1 upregulation may result from immune selection pressure induced by ICB or ACT therapies. To investigate this, we evaluated E2F1 expression in TC-1 and A375 cells across

successive rounds of immune selection (P0 to P3) using ICB and ACT, respectively. We observed a stepwise increase in E2F1 expression from P0 to P3 cells (Supplementary Fig. 14), suggesting that the elevated E2F1 levels in P3 cells likely result from the enrichment of E2F1⁺ cells during the immune selection process. Collectively, our findings suggest that immune pressure induced by immunotherapy drives the selection of E2F1⁺ tumor cells, which subsequently upregulate HMGCRCR expression and contribute to ferroptosis resistance.

[On page 15, Discussion section of the revised manuscript]

The upregulation of HMGCRCR has been reported in multiple tumors^{56,57}. However, the regulatory mechanisms of HMGCRCR expression, particularly in the course of CTL-mediated immune selection, has not yet been extensively studied. In this study, we report E2F1 as an upstream regulator of HMGCRCR, which is responsible for ferroptosis resistance of immune-edited tumor cells. Interestingly, the E2F1-binding element in the HMGCRCR promoter region is evolutionally conserved, suggesting the important role of E2F1 in HMGCRCR regulation. **Indeed, compared to their parental P0 cells, E2F1 is upregulated in immune-edited P3 cells, and knockdown of E2F1 in P3 cells reduces HMGCRCR levels and increases their susceptibility to ferroptosis.** While we have established that the E2F1-HMGCRCR axis drives ferroptosis resistance, this raises an important question: how is E2F1 upregulated during cancer immunoediting? In this regard, our previous studies demonstrated that CTL-mediated immune selection enriches immune-refractory cancer cells with CSC-like properties, driven by E2F1 through FGFR signaling or NANOG induction. NANOG further amplifies E2F1 activity via AKT-mediated RB suppression, forming a reciprocal loop^{42,43}. In light of this, our current study suggest that immune pressure imposed by immunotherapy drives the selection of E2F1⁺ tumor cells, which in turn upregulate HMGCRCR expression and promotes ferroptosis resistance.

3. The study demonstrates the efficacy of combining HMGCRCR inhibition with immunotherapy in mouse models. However, translating these findings to human patients requires careful consideration of potential side effects and the pharmacodynamics of HMGCRCR inhibitors in humans. Currently, there are no data provided by the authors on the toxicity effects in these animals.

We thank the reviewer for this insightful comment. To evaluate the potential toxicity of simvastatin in our experimental setting, we monitored the body weight of mice across all treatment groups and observed no significant differences (Revised Supplementary Figure 20). In addition, we assessed potential hepatotoxicity by measuring serum liver enzyme levels (AST/ALT), which showed no significant changes compared to the untreated control group. These findings suggest that the dose of simvastatin used in this study does not induce systemic or hepatic toxicity. Consistent with our observation, previous studies have reported that statins are safe for treating dyslipidemia and may even offer protective effects against liver injury (Huan Liang et al., Biomed Pharmacother, 2018; Rossana M. Calderon et al., Mayo Clin Proc, 2010). However, further studies are warranted to comprehensively evaluate the safety and pharmacodynamic profiles of HMGCRCR-targeting agents in clinical settings. We

have addressed this point in the revised Results sections and have included relevant references to support this discussion.

[On Supplementary Fig. 20a-20c of the revised manuscript]

Supplementary Figure 20

[On page 13, Results section of the revised manuscript]

To evaluate potential safety concerns regarding the application of HMGCR inhibitors in treatment, we monitored body weight across all treatment groups and found no significant differences (Supplementary Fig. 20a). After two weeks of treatment, simvastatin-treated mice showed a transient elevation in ALT and AST levels compared to the ICB monotherapy group, which returned to baseline within two weeks after treatment cessation (Supplementary Fig. 20b and 20c), consistent with previous clinical observations^{49,50}. These results suggest that the simvastatin dose used in this study does not induce systemic or hepatic toxicity. Taken together, our data indicate that HMGCR inhibition with simvastatin can reverse ferroptosis resistance of immune-edited tumor cells, thereby enhancing the therapeutic efficacy of T-cell based immunotherapy, such as ACT and ICB therapy.

4. The study focuses on the E2F1-HMGCR axis but does not extensively explore other potential mechanisms of ferroptosis resistance that might be relevant in different contexts or tumor types. The authors need to examine other ferroptosis resistance mechanisms, such as FSP1. It is also surprising that GPX4 is not changed in this study, as it is known that the mevalonate pathway can affect GPX4.

We thank the reviewer for this critical comment. As pointed out, GPX4 and FSP1 are well-established key regulators of ferroptosis resistance. In light of this, we carefully examined the expression levels of GPX4 and FSP1 in our experimental models. In both immune-refractory tumor cells (P3) and parental cells (P0), we observed no significant changes in GPX4 mRNA or protein levels, nor in GPX4 enzymatic activity (Revised Supplementary Fig 3 and Revised Supplementary Fig 8). Although the mevalonate pathway is known to regulate GPX4 activity, inhibition of HMGCR using either simvastatin or siHMGCR did not affect GPX4 activity in P3 cells (Revised Supplementary Fig 5 and Revised Supplementary Fig 17). This is consistent with previous study (Yao et al., J Nanobiotechnol, 2021), which reported that GPX4 expression was not significantly affected by simvastatin treatment.

These findings suggest that ferroptosis resistance in our model is mediated by alternative components of the mevalonate pathway, rather than by GPX4 regulation. Notably, in immune-refractory tumor cells (P3), CoQ10 levels are elevated, and similar to simvastatin treatment, inhibition of the CoQ10 pathway reverses ferroptosis resistance in P3 cells. Simvastatin treatment reduced CoQ10 production, thereby sensitizing P3 cells to ferroptosis. Furthermore, supplementation with CoQ10 reversed the increased ferroptosis sensitivity induced by simvastatin (Revised Fig 6e-6j). These findings highlight the critical role of the HMGCR-CoQ10 axis in regulating ferroptosis resistance in immune-refractory tumor cells.

In the case of FSP1, we did not observe any differences in gene expression in our immune-refractory tumor models (Revised Fig. 1e). For these reasons, we excluded the role of FSP1 in our study. However, since FSP1 is known to regulate ferroptosis resistance by reducing CoQ10, inhibitors targeting FSP1 could serve as a promising strategy to overcome ferroptosis resistance in immune-refractory tumors. These mechanistic clarifications have been incorporated into the revised manuscript to strengthen our discussion.

[On Supplementary Fig. 3 and 8 of the revised manuscript]

Supplementary Figure 3

Supplementary Figure 8

[On Supplementary Fig. 5 and 17 of the revised manuscript]

Supplementary Figure 5

Supplementary Figure 17

[On Fig. 6e-6j] of the revised manuscript]

Fig. 6

[On Fig. 1e of the revised manuscript]

Fig. 1

[On page 6, Results section of the revised manuscript]

Among the 17 genes responsible for the negative regulation of ferroptosis^{16,36}, we noted that the HMGR gene was upregulated in both B16 and TC-1 P3 cells compared to their P0 cells (Fig. 1f). Indeed, P3 cells had higher HMGR protein level and activity compared with P0 cells (Fig. 1g and 1h), without changes of the protein levels and activity of GPX4 or the protein levels of SLC7A11 (Supplementary Fig 3a and 3b), which are well-known regulators of ferroptosis resistance^{37,38}

[On page 6, Results section of the revised manuscript]

To directly link HMGR up-regulation to ferroptosis resistance, we silenced HMGR in B16 P3 and TC-1 P3 cells (Supplementary Fig 4a and 4b) and observed increased sensitivity of P3 cells to ferroptotic cell death induced by RSL3, which was accompanied by increased levels of lipid ROS (Fig. 1i,1j and Supplementary Fig 5), regardless of GPX4 activity.

[On page 8, Results section of the revised manuscript]

Moreover, we noted that levels of HMGR mRNA and protein were upregulated in A375 P3 cells compared to A375 P0 cells (Fig. 3d and 3e), without changes of the expression of GPX4 and SLC7A11 (Supplementary Figure 8a and 8b).

[On page 11, Results section of the revised manuscript]

These results suggest that simvastatin is an effective drug to reverse the ferroptosis resistance in E2F1^{high} immune refractory tumor cells. We next investigated the downstream pathways of HMGR that contribute to ferroptosis resistance. The mevalonate pathway is known to play a critical role in GPX4 maturation by supplying isopentenyl pyrophosphate (IPP)⁴⁶. However, we observed no changes in GPX4 expression or enzymatic activity in P3 cells compared to P0 cells (Supplementary Fig. 3a-3c and 8a-8b), or following HMGR inhibition by either simvastatin or siHMGR (Supplementary Fig. 5 and 17a-17b), suggesting that HMGR upregulation may confer ferroptosis resistance in immune-refractory tumors via GPX4-independent branches of the mevalonate pathway. Instead, we found that immune-refractory P3 tumor cells exhibited elevated levels of CoQ10 and cholesterol (Fig. 6e and 6f).

Interestingly, the ferroptosis-refractory phenotype of A375 P3 cells was reversed by treatment with 4-nitrobenzoate (4-NB), a CoQ10 synthesis inhibitor, but not by zaragozic acid (ZA), an inhibitor of squalene synthase involved in cholesterol synthesis (Fig. 6g and 6h), indicating that CoQ10, rather than cholesterol, plays a critical role in HMGCR-mediated ferroptosis resistance. Notably, simvastatin treatment did reduce CoQ10 levels in P3 cells, and supplementation with CoQ10 reversed the enhanced ferroptosis sensitivity induced by simvastatin (Fig. 6i and 6j). These findings underscore the pivotal role of the HMGCR–CoQ10 axis in regulating ferroptosis resistance in immune-refractory tumor cells.

[On page 14, Discussion section of the revised manuscript]

Given the crucial role of HMGCR in the mevalonate pathway, an important unanswered question is how this pathway contributes to ferroptosis resistance in tumor cells during cancer immunoediting. While it is well established that the mevalonate pathway supports ferroptosis resistance by supplying isopentenyl pyrophosphate (IPP) for GPX4 production⁴⁶, our study found no changes in GPX4 levels or activity in P3 cells or following HMGCR inhibition. This suggests that alternative mechanisms may underlie ferroptosis resistance. Notably, the FSP1–CoQ10 pathway has been identified as an independent mechanism from the classical GSH–GPX4 axis for suppressing lipid peroxidation and ferroptosis³⁶. In line with this, we observed elevated levels of CoQ10 and cholesterol in immune-refractory P3 tumor cells; however, only CoQ10 inhibition—not cholesterol inhibition—reversed ferroptosis resistance. These findings highlight CoQ10 as the key mediator of HMGCR-driven ferroptosis resistance. Thus, our data suggest that HMGCR may promote ferroptosis resistance in immune-edited tumor cells by enhancing CoQ10 biosynthesis. Although FSP1 expression remained unchanged in our models, its role in reducing CoQ10 to ubiquinol suggests that targeting FSP1, in addition to HMGCR, may help overcome ferroptosis resistance in immune-refractory tumors.

5. While the study shows increased infiltration of CD8⁺ T cells and enhanced anti-tumor immunity upon HMGCR inhibition, a more detailed analysis of the immune microenvironment (e.g., DAMP release, such as HMGB1) and other immune cell types (e.g., DC maturation and recruitment) could provide deeper insights into the overall immune response dynamics. Importantly, it needs to be confirmed whether the removal of CD8⁺ T cells will reverse the phenotype.

We thank the reviewer for this thoughtful recommendation. As suggested, we have further investigated the effects of HMGCR inhibition on DAMP release and DC recruitment in the TME. Our results showed that simvastatin-treated mice exhibited significantly higher serum levels of HMGB1 and an increased proportion of CRT⁺ tumor cells compared to the control group (revised Figure 7i and 7j), which was accompanied by increased conventional DC recruitment in tumor (revised Figure. 7k). Notably, all of these phenotypes were abolished by co-treatment with the ferroptosis inhibitor Lip1, confirming that DAMP release and DC recruitment were ferroptosis-dependent. In support, scRNA-seq

analysis revealed that immune-refractory P3 tumors showed reduced infiltration of CD8⁺ T cells and decreased cDC abundance compared to P0 tumors, suggesting an immunosuppressive microenvironment (Fig. 1, for reviewer only). These results indicate that ferroptosis-induced ICD may help restore immune activation by promoting DC maturation and subsequent T cell priming. Consistently, in vitro, co-treatment with simvastatin and RSL3 significantly elevated HMGB1 secretion and CRT surface exposure in B16 P3 and A375 P3 cells, whereas these effects were abolished by Lip-1 (revised Supplementary Fig. 19), confirming that HMGCR inhibition promotes ferroptosis-associated ICD.

Importantly, to confirm the role of CD8⁺ T cells in the observed therapeutic effect, we depleted CD8⁺ T cells using an anti-CD8 antibody. This depletion significantly diminished the therapeutic benefits of the simvastatin and anti-PD-1 combination treatment (Revised Supplementary Fig. 18), highlighting CD8⁺ T cells as key mediators of the anti-tumor efficacy of simvastatin combined with anti-PD-1 therapy. We have incorporated these new data and interpretations into the revised Results sections.

[Figure 1 for reviewer only]

[editorial note: figure redacted]

[On Fig. 7i-7k of the revised manuscript]

[On Supplementary Fig. 19 of the revised manuscript]

Supplementary Figure 19

a

b

c

[On Supplementary Fig. 18 of the revised manuscript]

Supplementary Figure 18

[On page 12, Results section of the revised manuscript]

Compared to treatment with anti-PD-1 or simvastatin alone, the combined therapy showed a remarkable therapeutic effect in B16 P3 tumor-bearing mice, which was attenuated by Lip-1 treatment (Fig. 7f). The combined treatment effect was accompanied by increased cell death and lipid ROS levels (Fig. 7g and 7h). **To confirm the role of CD8⁺ T cells in the observed therapeutic effect, we depleted them using an anti-CD8 antibody. This depletion markedly reduced the therapeutic benefits of the simvastatin and anti-PD-1 combination treatment (Supplementary Fig. 18), indicating that the enhanced immunotherapeutic effect of this combination strategy is largely dependent on CD8⁺ T cells.**

[On page 13, Results section of the revised manuscript]

Accumulating evidence suggests that tumor cell ferroptosis can promote antitumor immunity by releasing damage-associated molecular patterns (DAMPs)^{47,48}. In our study, combined treatment with simvastatin and either RSL3 or T cell supernatant increased HMGB1 secretion and CRT exposure in B16 P3 and A375 P3 cells (Supplementary Fig. 19). These effects were abolished by co-treatment with Lip1, indicating that the DAMP release was dependent on ferroptosis. Consistent with our *in vitro* findings, simvastatin-treated mice showed significantly higher serum levels of HMGB1 and a greater proportion of CRT⁺ tumor cells compared to controls (Fig. 7i and 7j), along with increased recruitment of conventional DCs to the tumor (Fig. 7k). These effects were abolished by co-treatment with the ferroptosis inhibitor Lip-1, confirming that DAMP release and DC recruitment were dependent on ferroptosis (Fig. 7i–7k). Furthermore, the numbers of infiltrated functional CD8⁺ T cells were higher in dual therapy groups than in monotherapy groups (Fig. 7l and 7m).

6. Non-cell-type selective induction of ferroptosis can dampen CTL activity. It remains unknown how targeting the HMGCR pathway only induces ferroptosis in cancer cells, rather than in CD8⁺ T cells.

We understand the reviewer's concern and have added further discussion in the revised manuscript addressing the complex and controversial role of ferroptosis in modulating anti-tumor immune responses within the tumor microenvironment. The reviewer raises an important point regarding impact of HMGCR inhibition in a tumor microenvironment (TME). Statins targeting the HMGCR pathway exert pleiotropic effects depending not only on the cell type, but also on the type of statin (Teodor-D Brumeanu et al., Clin Immunol., 2006). Recent studies show that among all statins, simvastatin and pitavastatin suppressed highly immunogenic cancer with increasing the percentage of CD8⁺ T cells (Wanqiong Yuan et al., Biomed Pharmacother., 2022, Vikash Kansal et al., J Immunother Cancer., 2023, Jong Ho Park et al., Nat Commun., 2024). In addition, simvastatin induces CD8⁺ T cell activation by upregulating the expression of NFATc2 that plays a crucial role in regulating the expression of genes involved in T-cell activation and differentiation (Wanqiong Yuan et al., Biomed Pharmacother., 2022). Consistently, ferroptosis induction by simvastatin with α PD-1 treatment reduced tumor growth, which in turn indirectly increased CD8⁺ T cell activity, supporting that simvastatin enhances CD8⁺ T cell responses. Nonetheless, we do not exclude the possibility that HMGCR inhibition could affect the

infiltration of other immune cells, and that will be an interesting challenge for future study. We appreciate a helpful comment of the reviewer.

[On page 17, Discussion section of the revised manuscript]

It is important to note that the role of ferroptosis in promoting anti-tumor immune responses remains complex and controversial. While ferroptotic cancer cell death can trigger the release of damage-associated molecular patterns (DAMPs)—a hallmark of immunogenic cell death (ICD)⁶²—and has been proposed as a strategy to enhance cancer immunotherapy^{47,48}, its immunogenic potential has shown variability across experimental models⁶³. Moreover, recent studies underscore the dual role of ferroptosis in the tumor microenvironment: although ferroptosis in tumor cells may stimulate anti-tumor immunity, ferroptosis in immune cells such as CD8⁺ T cells, NK cells, and neutrophils can suppress it⁶⁴⁻⁶⁶. In this context, our findings demonstrate that targeting HMGCR in immune-refractory tumor cells induces ferroptosis-associated ICD and enhances anti-tumor immune activation during T-cell based immunotherapy. While we cannot completely exclude the possibility that HMGCR inhibition may influence other immune cell populations, the observed increase in CD8⁺ T cell activity suggests that simvastatin primarily promotes tumor cell-specific ferroptotic death without compromising CD8⁺ T cell function. Nonetheless, the successful clinical translation of HMGCR inhibitors as immunotherapy adjuvants will require a deeper mechanistic understanding of how to selectively harness ferroptosis to stimulate anti-tumor immunity while minimizing unintended effects on immune effector cells.

7. All in vivo studies used only one siRNA; other siRNAs need to be repeated to confirm their findings.

We understand the reviewer's concern regarding potential off-target effects due to the use of a single siRNA in our in vivo experiments. To address this, we validated our findings using an additional Hmgcr-targeting siRNA (siHmgcr#2). Consistent with our previous results using Hmgcr #1-CNP (revised Fig. 2b), siHmgcr#2-CNPs combined with anti-PD-1 therapy profoundly retarded the tumor growth, while the anti-PD-1 antibody alone did not affect. Importantly, this therapeutic effect was abrogated by co-treatment with the Lip-1, demonstrating that the anti-tumor response was dependent on ferroptosis induction (revised Supplementary Fig. 6). These results support the robustness and specificity of our findings and indicate that they are not due to off-target effects of a single siRNA. We have incorporated this new results in the revised Methods and Results section.

[On Supplementary Fig. 6 of the revised manuscript]

Supplementary Figure 6

[On page 6, Results section of the revised manuscript]

we reasoned that silencing of HMGR expression could reverse the resistance of ICB-refractory tumor to anti-PD-1 therapy. We treated B16 P3 tumor-bearing mice with an anti-PD-1 antibody along with chitosan nanoparticles (CNPs) carrying *siHmgcr* or *siGFP* (Fig. 2a and Supplementary Fig. 6). While the anti-PD-1 antibody alone did not affect the tumor growth, its combination with *siHmgcr* #1 or #2-CNPs profoundly retarded the tumor growth, which was reversed by the simultaneous Lip-1 treatment (Fig. 2b and Supplementary Fig. 6).

8. The clinical data assay did not suggest a role for ferroptosis or lipid peroxidation. The number of patient samples used is low.

We appreciate the reviewer's valuable comment. To address this limitation, we are currently expanding our clinical cohorts, including melanoma cases, and conducting ferroptosis marker analyses. Preliminary analysis of a previously published dataset (Gabriel Abril-Rodriguez et al., Nature Cancer, 2020) revealed that ferroptosis-associated gene signatures, along with other regulated cell death markers, were enriched in responders to anti-PD-1 therapy, but not in non-responders (Figure 2, for reviewer only). Although the dataset did not include explicit ferroptosis directionality or lipid peroxidation signature, and the available clinical annotation was limited, suggests a potential association between ferroptosis and ICB efficacy. Accordingly, we have added a new paragraph in the revised Discussion section to address the study's limitations, including the small number of patient samples and the lack of direct clinical evidence for ferroptosis or lipid peroxidation. This addition highlights the need for future studies involving larger cohorts and ferroptosis marker analyses to validate the clinical relevance of our findings. We have included the new information in the revised Discussion section

[Figure 2 for reviewer only]

[editorial note: figure redacted]

[On page 18, Discussion section of the revised manuscript]

Study limitations in our work include the small number of clinical samples, which restricted our ability to fully investigate the relationship among the E2F1–HMGCRCR axis, ferroptosis resistance, and ICB response in patient tumors. Our study provides mechanistic insight into how T cell–based immunotherapies, such as PD-1 blockade and adoptive T cell transfer, promote ferroptosis resistance through immunoediting in tumor cells. By identifying the E2F1–HMGCRCR axis as a key driver of this resistance, we demonstrate that HMGCRCR inhibition restores ferroptosis sensitivity, rendering immune-refractory tumor cells susceptible to ACT and PD-1 blockade.

However, due to the limited clinical samples, we were unable to directly assess ferroptosis-related features, such as lipid peroxidation, in patient specimens. Future studies involving larger patient cohorts and immunohistochemical analysis of ferroptosis markers will be essential to validate the clinical relevance of our findings. In addition, single-cell transcriptomic profiling of immune-refractory tumor models and patient samples will provide high-resolution insights into cell-type-specific HMGCRCR functions and regulatory networks. This approach is expected to refine our understanding of ferroptosis resistance and reveal complex intercellular interactions within immune-refractory tumors, ultimately informing precision immunotherapeutic strategies. Nonetheless, our findings provide a strong preclinical foundation for targeting ferroptosis resistance to improve the efficacy of T cell–based cancer immunotherapies.

Reviewer #2 (Remarks to the Author): with expertise in cancer immunology

In this manuscript, Son et al. investigated whether and how ferroptosis plays a role in the anti-tumor activity of T cell-based immunotherapies by using immunotherapy-refractory tumor models. They showed that T cell-based immunotherapy drives the development of ferroptosis resistance of tumor cells. Furthermore, they showed that E2F1 was upregulated in immunotherapy-resistant cancer cells after immune editing, and it in turn bound to the promoter of the HMGCR gene to upregulate HMGCR, thereby contributing to the resistance to ferroptosis. Finally, they showed that HMGCR inhibition by siRNA or simvastatin rendered immune-refractory tumor cells susceptible to T cell therapy and PD-1 blockade therapy. Overall, this study provided a new mechanism by which cancer cells deregulate ferroptosis to escape immunotherapy-induced antitumor immunity.

Specific issues include:

1. Although ferroptosis is getting more attention in the cancer therapy field, T cell-induced cancer cell apoptosis are still considered the major mechanism of immune surveillance. Can the authors use some markers that specific for ferroptosis to demonstrate that the majority of cancer cells are undergoing ferroptosis or apoptosis during immune attack?

We thank the reviewer for the insightful comment. To evaluate the major form of tumor cell death under CTL-mediated attack, we treated P0 and P3 cells with CTL-derived supernatants containing cytokines such as IFN γ previously shown to promote tumor cell death including ferroptosis and apoptosis (Wang, W. et al., Nature., 2019, Dragica et al., 2020, Biomarker Research, Hideo Komita., 2006, Journal of Hepatology), in the presence of the ferroptosis inhibitor Lip-1 or the pan-caspase inhibitor zVAD. Apoptosis was assessed using active caspase-3 as a marker, and ferroptosis was evaluated by measuring lipid ROS accumulation, widely accepted indicators of their respective cell death pathways. In these experiments, immune-refractory P3 cells displayed marked resistance to both apoptosis and ferroptosis (revised Supplementary Fig. 2 and 7). Importantly, in P0 cells, blocking either ferroptosis with Lip-1 or apoptosis with zVAD partially reduced CTL supernatant-induced tumor cell death, supporting the notion that both pathways contribute to CTL-mediated cytotoxicity. Moreover, RSL3-induced tumor cell death was almost completely prevented by Lip-1 but unaffected by zVAD (revised Fig. 1c and 1d), demonstrating that RSL3 specifically triggers ferroptosis. Together, these findings underscore that both apoptosis and ferroptosis contribute to CTL-mediated tumor cell killing and support our hypothesis that T cell-based immunotherapy-driven immune editing confers tumor cell resistance to ferroptosis upon cytotoxic T cell attack. These clarifications have been incorporated into the revised Results section for improved clarity.

[On Supplementary Fig. 2 and Supplementary Fig. 7 of the revised manuscript]

Supplementary Figure 2

Supplementary Figure 7

[On Fig. 1c, 1d of the revised manuscript]

Fig. 1

[On page 5, Results section of the revised manuscript]

Given ferroptosis, besides apoptosis, being a key anti-tumor mechanism promoted by CTL during ICB therapy¹⁴, we asked whether tumor cells acquire resistance to ferroptosis through ICB-mediated immune editing. In this regard, activated CTL-derived supernatants have been reported to promote both apoptosis and ferroptosis^{14,33,34}. To test this, we treated P0 and P3 cells with CTL-derived supernatants and assessed ferroptosis (characterized by lipid peroxidation) and apoptosis (marked by active caspase-3). Notably, CTL-derived supernatants increased the frequency of active caspase-3+ cells and elevated lipid ROS levels in immune-susceptible P0 cells (Supplementary Fig. 2a and 2b). These effects were partially blocked by the apoptosis inhibitor zVAD or the ferroptosis inhibitor Lip-1 (Supplementary Fig. 2a and 2b), indicating that both apoptosis and ferroptosis contribute to CTL-mediated tumor cell killing. In contrast, immune-refractory P3 cells exhibited resistance to both forms of cell death (Supplementary Fig. 2a and 2b). In addition, we compared the susceptibility of P3 versus P0 cells to RSL3 and erastin, known ferroptosis inducers that work through inhibiting GPX4 and cystine/glutamate antiporter (xCT), respectively^{19,35}. Based on the IC₅₀ values, indicating the concentration of drugs that inhibits 50% of cell viability, we found that P3 cells were less sensitive to the ferroptosis inducers than P0 cells (Fig. 1a and 1b). Indeed, RSL3-induced death in P0 cells was accompanied by increased lipid ROS production, both of which were reversed by Lip-1 but not by zVAD (Fig. 1c and 1d), indicating that RSL3 selectively induces ferroptosis in P0 cells. However, P3 cells did not show a significant difference in the cell death or the production of lipid ROS upon treatment with RSL3 (Fig. 1c and 1d), indicating ferroptosis resistance. These findings suggest that anti-PD-1 therapy-mediated immune selection promotes tumor cell resistance to ferroptosis in response to cytotoxic T cell attacks.

[On page 8, Results section of the revised manuscript]

We established an ACT-refractory A375 P3 model from the parental A375 cells (A375 P0) through three rounds of *in vivo* selection by HL-A2-restricted NYESO1-specific CD8⁺ T clone (MAK #11)⁴⁰. Intriguingly, like the ICB-refractory mouse tumor models, CTL-refractory human A375 P3 cells were more resistant to ferroptosis induced by either CTL-derived supernatant or RSL3 treatment than A375 P0 cells (Fig. 3a and Supplementary Figure 7a, 7b).

2. For Figure 1, did the authors detect differences in sensitivity to different death forms between P3 and P0 cell lines? Are there differences in other forms of cell death besides apoptosis and ferroptosis? Furthermore, did the authors assess the baseline expression levels of GPX4 and SLC7A11, which are critical factors controlling ferroptosis, between the P0 and P3 cell lines?

We thank the reviewer for the comment. As shown in the revised Supplementary Fig. 2 and Fig. 7 and mentioned question number 1, immune-refractory P3 cells were resistant to both CTL-mediated apoptosis and ferroptosis compared to their parental P0 counterparts. Beyond ferroptosis and apoptosis, we explored other form of regulated cell death. Analysis of a published melanoma patient

cohort (Gabriel Abril-Rodriguez et al., Nature Cancer, 2020) demonstrated that apoptosis, ferroptosis, necroptosis, and pyroptosis signatures were enriched in responders to anti-PD-1 therapy, but not in non-responders (Figure 2, for reviewer only). These findings suggest that immune-refractory tumors may acquire broad resistance to multiple forms of immunogenic cell death. We are currently preparing a separate manuscript to further investigate the mechanisms underlying various type of cell death resistance during immunotherapy.

In response to your suggestion, we compared GPX4 and SLC7A11 expression between P0 and P3 cells (revised Supplementary Fig. 3 and Fig. 8) and found no significant differences, indicating that these canonical regulators are not responsible for ferroptosis resistance in our model. Although the mevalonate pathway typically supports ferroptosis resistance via GPX4 synthesis, we observed no changes in GPX4 levels or activity in P3 cells or after HMGCR inhibition. Instead, we identified CoQ10 as the key mediator of HMGCR-driven ferroptosis resistance, highlighting the importance of the HMGCR-CoQ10 axis independent of GPX4 (revised Fig. 6e-6j). We have added these findings to the revised Results section. We thank the reviewer for raising this important point, which has helped us broaden the scope of our analysis.

[Figure 2, for reviewer only]

[editorial note: figure redacted]

[On Fig. 3 and 8 of the revised manuscript]

Supplementary Figure 3

Supplementary Figure 8

[On Fig. 6e-6j of the revised manuscript]

Fig. 6

[On page 6, Results section of the revised manuscript]

Among the 17 genes responsible for the negative regulation of ferroptosis^{16,36}, we noted that the HMGCR gene was upregulated in both B16 and TC-1 P3 cells compared to their P0 cells (Fig. 1f). Indeed, P3 cells had higher HMGCR protein level and activity compared with P0 cells (Fig. 1g and 1h), without changes of the protein levels and activity of GPX4 or the protein levels of SLC7A11 (Supplementary Fig 3a and 3b), which are well-known regulators of ferroptosis resistance^{37,38}

[On page 8, Results section of the revised manuscript]

Moreover, we noted that levels of HMGCR mRNA and protein were upregulated in A375 P3 cells compared to A375 P0 cells (Fig. 3d and 3e), without changes of the expression of GPX4 and SLC7A11 (Supplementary Figure 8a and 8b).

[On page 11, Results section of the revised manuscript]

These results suggest that simvastatin is an effective drug to reverse the ferroptosis resistance in E2F1^{high} immune refractory tumor cells. We next investigated the downstream pathways of HMGCR that contribute to ferroptosis resistance. The mevalonate pathway is known to play a critical role in GPX4 maturation by supplying isopentenyl pyrophosphate (IPP)⁴⁶. However, we observed no changes in GPX4 expression or enzymatic activity in P3 cells compared to P0 cells (Supplementary Fig. 3a-3c and 8a-8b), or following HMGCR inhibition by either simvastatin or siHMGCR (Supplementary Fig. 5 and 17a-17b), suggesting that HMGCR upregulation may confer ferroptosis resistance in immune-refractory tumors via GPX4-independent branches of the mevalonate pathway. Instead, we found that immune-refractory P3 tumor cells exhibited elevated levels of CoQ10 and cholesterol (Fig. 6e and 6f). Interestingly, the ferroptosis-refractory phenotype of A375 P3 cells was reversed by treatment with 4-nitrobenzoate (4-NB), a CoQ10 synthesis inhibitor, but not by zaragozic acid (ZA), an inhibitor of squalene synthase involved in cholesterol synthesis (Fig. 6g and 6h), indicating that CoQ10, rather than cholesterol, plays a critical role in HMGCR-mediated ferroptosis resistance. Notably, simvastatin treatment did reduce CoQ10 levels in P3 cells, and supplementation with CoQ10 reversed the enhanced ferroptosis sensitivity induced by simvastatin (Fig. 6i and 6j). These findings underscore the pivotal role of the HMGCR–CoQ10 axis in regulating ferroptosis resistance in immune-refractory tumor cells.

3. E2F1 has been involved in many important cellular functions including cell cycle progression and apoptosis. How could the authors exclude the possibility that E2F1 contributes to cancer cell resistance to immunotherapy by regulating cell cycle or apoptosis when under immune attack? Or by upregulating Nanog as in their previous publications (References 45,46)? Furthermore, it is still not answered that how E2F1 was upregulated from P0 to P3?

The reviewer raised an important point regarding upregulation of E2F1 in immune-refractory tumors. In our previous studies, we demonstrated that CTL-mediated immune selection promotes the emergence of cancer cells with enhanced survival advantages, leading to the enrichment of immune-

refractory cancer cells characterized by unique features, including cancer stem cell (CSC)-like properties. During this process, E2F1 may contribute to CSC-like traits by activating FGFR signaling or inducing the expression of the stemness factor NANOG, and is further amplified through a reciprocal regulatory loop in which NANOG-driven AKT hyperactivation impairs RB function, thereby enhancing E2F1 activity (Song KH et al., *Oncogenesis.*, 2017, Oh SJ et al., *Cancer Res.* 2018). Based on this, we hypothesized that the upregulation of E2F1 could be a consequence of immune selection pressure induced by ICB or ACT therapies. To investigate this, we evaluated E2F1 expression in TC-1 and A375 cells across successive rounds of immune selection (P0 to P3) using ICB and ACT, respectively. We observed a stepwise increase in E2F1 expression from P0 to P3 cells. These findings suggest that the elevated E2F1 levels in P3 cells are likely the result of enrichment of E2F1+ cells during the immune selection process (revised Supplementary Figure 14). We have incorporated these clarifications into the revised Results and Discussion sections.

[On Supplementary Fig. 14 of the revised manuscript]

Supplementary Figure 14. Son et al

[On page 10, Results section of the revised manuscript]

We next investigated the underlying mechanism responsible for the upregulation of E2F1 during immune editing. Our previous study showed that CTL-mediated immune selection enriches immune-refractory cancer cells with cancer stem cell (CSC)-like properties^{9,42}. In this regard, E2F1 contributes to CSC-like properties by activating FGFR signaling or inducing the stemness factor NANOG, and is further amplified through a reciprocal regulatory loop in which NANOG-driven AKT hyperactivation impairs RB function, thereby enhancing E2F1 activity^{42,43}. Based on this, we hypothesized that E2F1 upregulation may result from immune selection pressure induced by ICB or

ACT therapies. To investigate this, we evaluated E2F1 expression in TC-1 and A375 cells across successive rounds of immune selection (P0 to P3) using ICB and ACT, respectively. We observed a stepwise increase in E2F1 expression from P0 to P3 cells (Supplementary Fig. 14), suggesting that the elevated E2F1 levels in P3 cells likely result from the enrichment of E2F1⁺ cells during the immune selection process. Collectively, our findings suggest that immune pressure induced by immunotherapy drives the selection of E2F1⁺ tumor cells, which subsequently upregulate HMGCR expression and contribute to ferroptosis resistance.

[On page 15, Discussion section of the revised manuscript]

Interestingly, the E2F1-binding element in the HMGCR promoter region is evolutionally conserved, suggesting the important role of E2F1 in HMGCR regulation. Indeed, compared to their parental P0 cells, E2F1 is upregulated in immune-edited P3 cells, and knockdown of E2F1 in P3 cells reduces HMGCR levels and increases their susceptibility to ferroptosis. While we have established that the E2F1-HMGCR axis drives ferroptosis resistance, this raises an important question: how is E2F1 upregulated during cancer immunoediting? In this regard, our previous studies demonstrated that CTL-mediated immune selection enriches immune-refractory cancer cells with CSC-like properties, driven by E2F1 through FGFR signaling or NANOG induction. NANOG further amplifies E2F1 activity via AKT-mediated RB suppression, forming a reciprocal loop^{42,43}. In light of this, our current study suggest that immune pressure imposed by immunotherapy drives the selection of E2F1⁺ tumor cells, which in turn upregulate HMGCR expression and promotes ferroptosis resistance.

4. The downstream mechanism by which HMGCR downregulates ferroptosis sensitivity remain unclear. The authors only showed some preliminary data in Supplementary Figure S8 by using two chemical inhibitors. This is an important part of the study, and I would suggest put these results in a main figure, and rigorously perform CoQ10 and Cholesterol supplement and depletion experiment to verify their role in HMGCR-mediated ferroptosis resistance.

We thank the reviewer for this helpful suggestion. To address this issue, we have performed additional experiments to clarify the role of CoQ10 and cholesterol in HMGCR-mediated ferroptosis resistance. Our studies reveal that the level of CoQ10, generated through the mevalonate pathway, was increased in immune-refractory tumor cells and We confirmed that CoQ10 production is reduced by treatment with HMGCR inhibitor, simvastatin (revised Fig. 6f, 6i). To assess functional relevance, we treated P3 cells with both simvastatin and RSL3, followed by CoQ10 supplementation. We observed that the addition of CoQ10 prevented the induction of ferroptosis (revised Fig. 6j). Therefore, the HMGCR-CoQ10 pathway is upregulated in immune-refractory tumor cells, thereby inducing ferroptosis resistance. In response to the reviewer's recommendation, we have moved these results from Supplementary data (Supplementary Figure S8) to the main figures (revised Fig. 6g and 6h) and incorporated the findings into the revised Results, Methods, and Figure Legends sections. We thank the reviewer for this helpful suggestion, which has strengthened the mechanistic framework of our study.

[On Fig 6e-6j] of the revised manuscript]

Fig. 6

[On page 11, Results section of the revised manuscript]

These results suggest that simvastatin is an effective drug to reverse the ferroptosis resistance in E2F1^{high} immune refractory tumor cells.

We next investigated the downstream pathways of HMGR that contribute to ferroptosis resistance. The mevalonate pathway is known to play a critical role in GPX4 maturation by supplying isopentenyl pyrophosphate (IPP)⁴⁶. However, we observed no changes in GPX4 expression or enzymatic activity in P3 cells compared to P0 cells (Supplementary Fig. 3a-3c and 8a-8b), or following HMGR inhibition by either simvastatin or siHMGR (Supplementary Fig. 5 and 17a-17b), suggesting that HMGR upregulation may confer ferroptosis resistance in immune-refractory tumors via GPX4-independent branches of the mevalonate pathway. Instead, we found that immune-refractory P3 tumor cells exhibited elevated levels of CoQ10 and cholesterol (Fig. 6e and 6f). Interestingly, the ferroptosis-refractory phenotype of A375 P3 cells was reversed by treatment with 4-nitrobenzoate (4-NB), a CoQ10 synthesis inhibitor, but not by zaragozic acid (ZA), an inhibitor of squalene synthase involved in cholesterol synthesis (Fig. 6g and 6h), indicating that CoQ10, rather than cholesterol, plays a critical role in HMGR-mediated ferroptosis resistance. Notably, simvastatin treatment did reduce CoQ10 levels in P3 cells, and supplementation with CoQ10 reversed the enhanced ferroptosis sensitivity induced by simvastatin (Fig. 6i and 6j). These findings underscore the pivotal role of the HMGR–CoQ10 axis in regulating ferroptosis resistance in immune-refractory tumor cells.

5. In Figure 2, the tumors in siHMGR were significantly reduced compared with those in siGFP group. What was the underlying reason for tumor regression by HMGR knockdown? Could lipro-1 rescue this part of the tumor regression?

We thank the reviewer for raising this important concern. While HMGR inhibition is known to promote ferroptosis under certain conditions, our data showed that HMGR knockdown alone did not strongly induce classical features of ferroptosis, such as increased 7-AAD⁺ cells and lipid ROS (Fig. 2c and d). However, when combined with PD-1 blockade treatment, we observed a marked increase in the percentage of 7AAD⁺ tumor cells and the lipid ROS level, both of which were reversed by concomitant Lip-1 treatment (Fig. 2c and d). These findings suggest that HMGR inhibition creates a cellular state that sensitizes immune-refractory tumor cells to ferroptosis, which is further activated in the context of T cell-mediated immune attack facilitated by PD-1 blockade.

However, we do not exclude the involvement of ferroptosis-independent mechanisms in the antitumor effects of HMGR inhibition. For example, disruption of protein prenylation, a downstream process of the mevalonate pathway, may impair oncogenic signaling and contribute to reduced tumor cell proliferation and survival (Patience Odeniyide et al., Oncogene., 2022, Wei-Hua Wang et al., Acta Pharmacol Sin., 2021). Consistent with this, lip-1 treatment in the siHMGR-only group did not reverse tumor regression (revised Supplementary figure 6b), suggesting that additional mechanisms beyond ferroptosis may contribute to the antitumor effects of HMGR knockdown. Therefore, the therapeutic effect of HMGR knockdown likely results from a combination of ferroptosis sensitization under immune pressure and additional anti-proliferative mechanisms, such as impaired protein prenylation. These findings highlight the potential of HMGR inhibition as a multifaceted strategy to enhance the efficacy of T cell-based immunotherapies. We have included the new information in the revised Discussion section

[On Supplementary Fig 6b of the revised manuscript]

Supplementary Figure 6

[On page 16, Discussion section of the revised manuscript]

It is intriguing, and perhaps counterintuitive, that HMGCR inhibition alone did not strongly induce classical features of ferroptosis, such as increased 7-AAD⁺ cells and lipid ROS. However, when combined with PD-1 blockade, it did trigger ferroptosis in immune-refractory tumor cells, suggesting that HMGCR inhibition primes tumor cells for ferroptosis, which is subsequently activated during T cell-mediated immune attack facilitated by PD-1 blockade. Nonetheless, we do not exclude the possibility that ferroptosis-independent mechanisms also contribute to the antitumor effects of HMGCR inhibition. For instance, disrupting protein prenylation, a downstream process of the mevalonate pathway, can impair oncogenic signaling, reduce tumor cell proliferation, and enhance immunogenicity by interfering with RAS or RAC1 function⁵⁹⁻⁶¹. Thus, the therapeutic benefits of HMGCR inhibition likely result from both ferroptosis sensitization under immune pressure and additional anti-proliferative or cytotoxic mechanisms, such as impaired prenylation, underscoring its multifaceted potential to enhance the efficacy of T cell-based immunotherapies.

6. HMGCR inhibition has been shown to increase cancer cell immunogenicity either by inhibiting RAS or RAC1 protein prenylation, and induce immunogenic cell death (e.g., PMID: 34330763, 34266895). Did the authors observe changes of expression of certain immunogenic molecules such as HMGB1 and calreticulin after HMGCR inhibition on cancer cells? Whether blockade of ferroptosis would decrease the expression of these immunogenic molecules?

We thank the reviewer for raising this important concern. To address this question, we have further investigated the effects of HMGCR inhibition on DAMP release in the TME. Our results showed that simvastatin-treated mice exhibited significantly higher serum levels of HMGB1 and an increased proportion of CRT⁺ tumor cells compared to the control group (Revised Figure 7i and 7j), which was accompanied by increased conventional DC recruitment in tumor (Revised Figure. 7k). These effects were abolished by co-treatment with the ferroptosis inhibitor Lip1, indicating that observed immunogenic changes including DAMP release and DC recruitment were ferroptosis-dependent (Revised Figure 7i-7k). Consistent with our *in vivo* findings, combined treatment with simvastatin and RSL3 *in vitro* induced robust release of HMGB1 and CRT translocation to the surface of immune-refractory tumor cells. These effects were abrogated by co-treatment with the ferroptosis inhibitor Lip-1 (Revised Supplementary Fig. 19). These findings suggest that HMGCR inhibition sensitizes immune-refractory tumor cells to ferroptosis-associated immunogenic cell death (ICD), thereby promoting anti-tumor immune activation. We have included this information in the revised Results and Discussion section.

[On Fig 7i-7k of the revised manuscript]

Fig. 7

[On Supplementary Fig 19 of the revised manuscript]

Supplementary Figure 19

[On page 12, Result section of the revised manuscript]

The combined treatment effect was accompanied by increased cell death and lipid ROS levels (Fig. 7g and 7h). To confirm the role of CD8⁺ T cells in the observed therapeutic effect, we depleted them using an anti-CD8 antibody. This depletion markedly reduced the therapeutic benefits of the simvastatin and anti-PD-1 combination treatment (Supplementary Fig. 18), indicating that the enhanced immunotherapeutic effect of this combination strategy is largely dependent on CD8⁺ T cells. Accumulating evidence suggests that tumor cell ferroptosis can promote antitumor immunity by releasing damage-associated molecular patterns (DAMPs)^{47,48}. In our study, combined treatment with simvastatin and either RSL3 or T cell supernatant increased HMGB1 secretion and CRT exposure in B16 P3 and A375 P3 cells (Supplementary Fig. 19). These effects were abolished by co-treatment with Lip1, indicating that the DAMP release was dependent on ferroptosis. Consistent with our *in vitro* findings, simvastatin-treated mice showed significantly higher serum levels of HMGB1 and a greater proportion of CRT⁺ tumor cells compared to controls (Fig. 7i and 7j), along with increased recruitment of conventional DCs to the tumor (Fig. 7k). These effects were abolished by co-treatment with the ferroptosis inhibitor Lip-1, confirming that DAMP release and DC recruitment were dependent on ferroptosis (Fig. 7i–7k). Furthermore, the numbers of infiltrated functional CD8⁺ T cells were higher in dual therapy groups than in monotherapy groups (Fig. 7l and 7m).

[On page 17, Discussion section of the revised manuscript]

It is important to note that the role of ferroptosis in promoting anti-tumor immune responses remains complex and controversial. While ferroptotic cancer cell death can trigger the release of damage-associated molecular patterns (DAMPs)—a hallmark of immunogenic cell death (ICD)⁶²—and has been proposed as a strategy to enhance cancer immunotherapy^{47,48}, its immunogenic potential has shown variability across experimental models⁶³. Moreover, recent studies underscore the dual role of ferroptosis in the tumor microenvironment: although ferroptosis in tumor cells may stimulate anti-tumor immunity, ferroptosis in immune cells such as CD8⁺ T cells, NK cells, and neutrophils can suppress it^{64–66}. In this context, our findings demonstrate that targeting HMGCR in immune-refractory tumor cells induces ferroptosis-associated ICD and enhances anti-tumor immune activation during T-cell based immunotherapy. While we cannot completely exclude the possibility that HMGCR inhibition may influence other immune cell populations, the observed increase in CD8⁺ T cell activity suggests that simvastatin primarily promotes tumor cell-specific ferroptotic death without compromising CD8⁺ T cell function. Nonetheless, the successful clinical translation of HMGCR inhibitors as immunotherapy adjuvants will require a deeper mechanistic understanding of how to selectively harness ferroptosis to stimulate anti-tumor immunity while minimizing unintended effects on immune effector cells.

7. In Figure 3f, after E2F1 knockdown, the promoter activity of HMGCRCR was basically abolished, but why was the expression of Hmgcr in Figure 3g still much higher than that in the control group after mutation of the binding site?

We thank the reviewer for raising this important concern. A detailed analysis of the HMGCRCR promoter identified two putative E2F1 binding sites (E1 and E2), both highly conserved across humans to mice (revised Fig. 4b). Mutation of either site led to a partial reduction in HMGCRCR promoter activity, while simultaneous mutation of both sites nearly abolished promoter activity (revised Fig. 4e and 4g), consistent with the effect observed following E2F1 knockdown. Furthermore, quantitative ChIP (qChIP) analysis using primers targeting the two adjacent binding elements confirmed that E2F1 directly binds to this region of the HMGCRCR promoter (Fig. 4h). We appreciate the reviewer's insightful feedback, which significantly improved the clarity of our mechanistic interpretation. We have included this information in the revised Results section.

[On Fig 4b, 4e and 4g of the revised manuscript]

[On page 09, Results section of the revised manuscript]

Interestingly, we identified two highly conserved putative E2F1-binding elements in the promoter regions of both human and mouse HMGCRCR genes, supporting the possibility that E2F1 functions as a direct transcriptional activator of HMGCRCR (Fig. 4b). Indeed, silencing of E2F1 in A375 P3 cells decreased both HMGCRCR protein and mRNA levels (Fig. 4c and 4d). We then engineered a reporter that expressed luciferase under the control of the human HMGCRCR promoter (pGL3-HMGCRCR pro) (Fig. 4e). Notably, the promoter activity was higher in A375 P3 cells than in A375 P0 cells; however, the increased promoter activity in A375 P3 cells was diminished by E2F1 depletion (Fig. 4f). In addition, mutation of either or both E2F1-binding sites in the HMGCRCR promoter decreased the luciferase activity in A375 P3 cells, suggesting that the E2F1-binding sites might act as cis-acting elements responsible for the basal activity of HMGCRCR promoter (Fig. 4e and 4g). Quantitative Chromatin immunoprecipitation (qChIP) assays using primers targeting the two adjacent E2F1-binding sites confirmed the direct binding of E2F1 to the putative binding region within the HMGCRCR promoter and showed that E2F1 binding was greater in P3 cells than in P0 cells (Fig. 4h). These findings demonstrate that E2F1 up-regulates HMGCRCR transcription by directly binding to its promoter region.

8. The cells used in Figure 6b and Figure 6e were both A375 P0 cells, why were the sensitivity to statin killing obviously different?

We understand the reviewer's concern and apologize for our mistakes. Upon reviewing our data, we found that the observed difference in statin sensitivity between Figure 6b and Figure 6e (revised Supplementary Fig. 16a) was likely due to variations in cell number during the initial seeding process, which can influence drug response. To avoid this confusion and clarify the data, we have repeated the experiment and replaced the data for sensitivity to simvastatin in A375 P0 cells (revised Figure 6b). We have included this data in the revised manuscripts.

[On Fig 6b of the revised manuscript]

Fig. 6

9. The internal parameters (actin) depicted in Figure 1g exhibited inconsistencies, and the upregulation of HMGCR was not very pronounced. Similarly, the upregulation of HMGCR1 expression in Figure 3e was also not obvious. Quantification of the Western Blot (WB) results may enhance the credibility of these results.

We thank the reviewer for this helpful suggestion. We have re-examined and performed quantification of the WB data, normalizing HMGCR expression levels to actin to ensure consistency in protein loading and improve the reliability of the results. In the revised manuscript, we have displayed representative Western blot images, and quantified the Western blot data as recommended.

[On Fig. 1g, 3e, 4c and supplementary Fig. 3a, 4a, 4b, 8a and 12a of the revised manuscript]

10. Minor: There are a few writing mistakes throughout the manuscript, for example, the IC50 unit in Figure 1 should be “ μM ”; Please double check.

We thank the reviewer for pointing out our mistake. As suggested, we have carefully reviewed the manuscript and corrected the IC50 unit in Figure 1, changing “ $\text{u}\mu\text{M}$ ” to “ μM ” in the Results section of the revised manuscript accordingly.

Reviewer #3 (Remarks to the Author): with expertise in cancer immunology

This is an interesting study shedding some light on the potential mechanism(s) of how statin drugs enhance responses to immunotherapy. The authors use B16 and TC-1 models, in addition to an adoptive cell transfer (ACT) mouse model, to show that immunotherapy resistance involves high expression of E2F1 and HMGCR leading to resistance to ferroptosis. Simvastatin or manipulation of E2F1/HMGCR are then able to restore ferroptosis, sensitivity to immunotherapy and CD8 T cell infiltration. Overall, this is a well performed study with appropriate controls, and it is overall well written.

Major concerns:

1) The Discussion is very limited. It includes a long summary of the results and potential implications, but there is little to no discussion of outside work and previously published studies for context. There are several preclinical and retrospective clinical studies showing that statins enhance responses to T-cell-based immunotherapy, and those should be cited here (and similarly the statement in the Introduction that "the impact of HMGCR on the resistance to anti-tumor immunity and ultimately to ICB is not yet known" is really not true). There is also no paragraph acknowledging limitations of the study.

We appreciate the reviewer's insightful comments. Accordingly, we have revised the Introduction and Discussion section to address the concern and added further discussion on the study's limitations in the revised Discussion section.

[On page 4, Introduction section of the revised manuscript]

Notably, inhibiting HMGCR with statins downregulates the mevalonate pathway and results in the induction of ferroptosis^{27,28}. **In addition, several preclinical and retrospective clinical studies have shown that statins could enhance the efficacy of T cell–based immunotherapy^{29,30}. Despite growing importance of HMGCR as a therapeutic target, the potential link between HMGCR and cancer immunoediting remains poorly understood.**

[On page 18, Discussion section of the revised manuscript]

Study limitations in our work include the small number of clinical samples, which restricted our ability to fully investigate the relationship among the E2F1–HMGCR axis, ferroptosis resistance, and ICB response in patient tumors. Our study provides mechanistic insight into how T cell–based immunotherapies, such as PD-1 blockade and adoptive T cell transfer, promote ferroptosis resistance through immunoediting in tumor cells. By identifying the E2F1–HMGCR axis as a key driver of this resistance, we demonstrate that HMGCR inhibition restores ferroptosis sensitivity, rendering immune-refractory tumor cells susceptible to ACT and PD-1 blockade.

However, due to the limited clinical samples, we were unable to directly assess ferroptosis-related features, such as lipid peroxidation, in patient specimens. Future studies involving larger patient cohorts and immunohistochemical analysis of ferroptosis markers will be essential to validate the clinical

relevance of our findings. In addition, single-cell transcriptomic profiling of immune-refractory tumor models and patient samples will provide high-resolution insights into cell-type-specific HMGCR functions and regulatory networks. This approach is expected to refine our understanding of ferroptosis resistance and reveal complex intercellular interactions within immune-refractory tumors, ultimately informing precision immunotherapeutic strategies. Nonetheless, our findings provide a strong preclinical foundation for targeting ferroptosis resistance to improve the efficacy of T cell–based cancer immunotherapies.

2) More info is needed on the methods used to obtain data for Figure 5. What kind of cancers did these patients have? How was the 30% cutoff determined was an ROC curve done? Does the Kaplan-Meier curve represent overall survival or progression-free survival?

We thank the reviewer for the thoughtful comments. The data in Figure 5 were obtained from a melanoma patient cohort, and the Kaplan–Meier curves represent overall survival (OS). Initially, cut-off values for E2F1 and HMGCR expression were based on percentile stratification. Following the reviewer’s suggestion, we performed ROC analysis using Youden’s J statistic to define optimal thresholds (E2F1: -0.882; HMGCR: -0.724; combined: -0.655), which improved the statistical significance of the survival analysis (revised Fig. 5c–5e). We thank the reviewer for this suggestion, which helped us refine our clinical data and improve the clarity and rigor of our results.

[Figure 3, for reviewer only]

[editorial note: figure redacted]

[On Fig 5c-5e of the revised manuscript]

Fig. 5

Minor concerns:

1) The authors claim that ferroptosis can induce immunogenic cell death, but the literature on this is mixed. ICD is not just any type of cell death that promotes immunity, but a distinct process involving specific DAMPs (see studies by Lorenzo Galluzzi for example). Citing one paper and saying that ferroptosis induces ICD is not quite accurate.

We thank the reviewer for this valuable comment. We agree that the role of ferroptosis in inducing immunogenic cell death (ICD) remains controversial, and that ICD requires specific DAMP release and immune activation, as emphasized in studies by Galluzzi and colleagues. In our study, we additionally performed experiments and observed HMGB1 release, calreticulin exposure, and cDC1 recruitment following HMGCRCR inhibition and ferroptosis induction (revised figure 7), suggesting potential immunogenic features. However, we acknowledge that these findings alone do not definitively establish ferroptosis as a bona fide ICD pathway. Accordingly, we have included additional discussion on the controversial role of ferroptosis in induction of ICD in the revised Discussion section.

[On page 17, Discussion section of the revised manuscript]

It is important to note that the role of ferroptosis in promoting anti-tumor immune responses remains complex and controversial. While ferroptotic cancer cell death can trigger the release of damage-associated molecular patterns (DAMPs)—a hallmark of immunogenic cell death (ICD)⁶²—and has been proposed as a strategy to enhance cancer immunotherapy^{47,48}, its immunogenic potential has shown variability across experimental models⁶³. Moreover, recent studies underscore the dual role of ferroptosis in the tumor microenvironment: although ferroptosis in tumor cells may stimulate anti-tumor immunity, ferroptosis in immune cells such as CD8⁺ T cells, NK cells, and neutrophils can suppress it⁶⁴⁻⁶⁶. In this context, our findings demonstrate that targeting HMGCRCR in immune-refractory tumor cells induces ferroptosis-associated ICD and enhances anti-tumor immune activation during T-cell based immunotherapy. While we cannot completely exclude the possibility that HMGCRCR inhibition may

influence other immune cell populations, the observed increase in CD8⁺ T cell activity suggests that simvastatin primarily promotes tumor cell-specific ferroptotic death without compromising CD8⁺ T cell function. Nonetheless, the successful clinical translation of HMGCR inhibitors as immunotherapy adjuvants will require a deeper mechanistic understanding of how to selectively harness ferroptosis to stimulate anti-tumor immunity while minimizing unintended effects on immune effector cells.

2) Typo on page 14: "T cell-base" should be "T cell-based"

We thank the reviewer for pointing out our mistake. As suggested, the word of "T cell-base" was changed to the word of "T cell-based" in the Discussion section of the revised manuscript.

[On page 17, Discussion section of the revised manuscript]

During **T cell-based** cancer immunotherapy, ferroptosis of tumor cells by activated T cells can contribute to a sustained antitumor effect by activating anti-tumor immunity

3) Are figures 6a, 6b and 6e also using trypan blue? This is worth mentioning again in the legend.

We thank the reviewer for this helpful suggestion. As suggested, we have mentioned the use of "the trypan blue exclusion method" in the figure legend of the revised manuscript.

[On page 41, Figure legend section of the revised manuscript]

Fig. 6 Simvastatin overcomes ferroptosis resistance in Immune-refractory tumors via Inhibition of the HMGCR–CoQ10 Axis. a and b Cell death percentage of B16 P0 and P3 cells **a** or A375 P0 and P3 cells **b** treated with indicated concentrations of RSL3 with or without Sim (1 μM) for 24h. **The percentage of cell death was determined by trypan blue exclusion assay.**

Reviewer #4 (Remarks to the Author): with expertise in ferroptosis, cancer immunology)

Son et al. studied the regulations of E2F1 and HMGCR on tumoral ferroptosis and immunotherapy resistance using both murine and human tumor and specific CTL models. They found that immunotherapy resistant tumor cells generated from in vivo ICB treatment became resistant to ferroptosis inducers in vitro. Mechanistically, these resistant cells express higher level of HMGCR and E2F1, and knockdown of these two genes could sensitize tumor to ferroptosis and ICB therapy. Meanwhile, they explored the regulatory mechanism underlying HMGCR transcription by E2F1. Simvastatin, as a HMGCR inhibitor, was shown to enhance tumoral ferroptosis and augment ACT- or ICB-mediated therapy. Overall, this study is well performed and the data is well organized. However, there are still several concerns need to be carefully addressed.

Major concerns:

1. The authors speculated that the CTL-resistant P3 cells became more resistance to ferroptosis, but how about apoptosis? The in vitro co culture of tumor and CD8 T cells is needed to test the difference between P0 and P3 cells against CTL cytotoxicity.

We thank the reviewer for the comment. In our previous publications, we have demonstrated through co-culture experiments with cognate CTLs that both ICB-resistant B16 P3 cells (Kim S et al., Cancer Immunol Res. 2025) and CTL-resistant A375 P3 cells (Lee HJ et al., Nat Commun. 2022) exhibit resistance to CTL-mediated apoptosis. In the current study, we further confirmed that ICB-resistant TC-1 P3 cells also show reduced sensitivity to CTL-mediated apoptosis (revised supplementary Fig. 1c), indicating that immune-refractory P3 cells exhibits resistance not only to ferroptosis but also to apoptosis in the context of T cell-mediated cytotoxicity.

To further delineate the contribution of ferroptosis and apoptosis resistance, we initially established a direct CTL-tumor cell co-culture system but found that rapid-onset apoptosis masked ferroptotic responses, which occur with slower kinetics. To overcome this, we treated P0 and P3 cells with CTL-derived supernatants enriched in cytokines such as IFN γ , which have been reported to promote tumor cell death, including ferroptosis and apoptosis (Wang et al., Nature, 2019; Dragica et al., Biomarker Research, 2020; Komita et al., Journal of Hepatology, 2006), in the presence of the ferroptosis inhibitor Lip-1 and the apoptosis inhibitor zVAD. Compared to immune-susceptible P0 tumor cells, immune-refractory P3 cells exhibited resistance to both types of cell death, as shown by reduced levels of active caspase-3⁺ cells and lower lipid ROS production (revised Supplementary Fig. 2a, 2b, and 7a, 7b). Notably, in P0 cells, treatment with either Lip-1 or zVAD partially blocked CTL supernatant-induced cell death, indicating that both apoptosis and ferroptosis contribute to CTL-mediated tumor cell killing. In addition, RSL3-induced cell death in P0 cells was almost fully suppressed by Lip-1 but remained unaffected by zVAD (revised Fig. 1c and 1d), confirming that RSL3 specifically triggers ferroptosis rather than apoptosis. Consistent with these in vitro findings, in vivo experiments where both direct and indirect T cell effects operative, showed that HMGCR inhibition (via siHMGCR or simvastatin) combined with anti-PD-1 therapy resulted in significant tumor control, which was reversed by Lip-1 treatment (revised Fig. 2 and 7). These findings reinforce our hypothesis that T cell-based immunotherapy-driven immune

editing promotes tumor cell resistance to ferroptosis during cytotoxic T cell-mediated immune responses
 We have clarified these experimental rationales and findings in the revised Results sections for greater transparency.

[On Supplementary Fig. 2 and Supplementary Fig. 7 of the revised manuscript]

Supplementary Figure 2

Supplementary Figure 7

[On Fig. 1c, 1d of the revised manuscript]

Fig. 1

[On page 5, Results section of the revised manuscript]

Given ferroptosis, besides apoptosis, being a key anti-tumor mechanism promoted by CTL during ICB therapy¹⁴, we asked whether tumor cells acquire resistance to ferroptosis through ICB-mediated immune editing. In this regard, activated CTL-derived supernatants have been reported to promote both apoptosis and ferroptosis^{14,33,34}. To test this, we treated P0 and P3 cells with CTL-derived supernatants and assessed ferroptosis (characterized by lipid peroxidation) and apoptosis (marked by active caspase-3). Notably, CTL-derived supernatants increased the frequency of active caspase-3+ cells and elevated lipid ROS levels in immune-susceptible P0 cells (Supplementary Fig. 2a and 2b). These effects were partially blocked by the apoptosis inhibitor zVAD or the ferroptosis inhibitor Lip-1 (Supplementary Fig. 2a and 2b), indicating that both apoptosis and ferroptosis contribute to CTL-mediated tumor cell killing. In contrast, immune-refractory P3 cells exhibited resistance to both forms of cell death (Supplementary Fig. 2a and 2b). In addition, we compared the susceptibility of P3 versus P0 cells to RSL3 and erastin, known ferroptosis inducers that work through inhibiting GPX4 and cystine/glutamate antiporter (xCT), respectively^{19,35}. Based on the IC₅₀ values, indicating the concentration of drugs that inhibits 50% of cell viability, we found that P3 cells were less sensitive to the ferroptosis inducers than P0 cells (Fig. 1a and 1b). Indeed, RSL3-induced death in P0 cells was accompanied by increased lipid ROS production, both of which were reversed by Lip-1 but not by zVAD (Fig. 1c and 1d), indicating that RSL3 selectively induces ferroptosis in P0 cells. However, P3 cells did not show a significant difference in the cell death or the production of lipid ROS upon treatment with RSL3 (Fig. 1c and 1d), indicating ferroptosis resistance. These findings suggest that anti-PD-1 therapy-mediated immune selection promotes tumor cell resistance to ferroptosis in response to cytotoxic T cell attacks.

[On page 8, Results section of the revised manuscript]

We established an ACT-refractory A375 P3 model from the parental A375 cells (A375 P0) through three rounds of *in vivo* selection by HL-A2-restricted NYESO1-specific CD8⁺ T clone (MAK #11)³⁸. Intriguingly, like the ICB-refractory mouse tumor models, CTL-refractory human A375 P3 cells were more resistant to ferroptosis induced by either CTL-derived supernatant or RSL3 treatment than A375 P0 cells (Fig. 3a and Supplementary Figure 7a, 7b).

2. HMGCR is still expressed in P0 cells, but why its knockdown has no effect on RSL3-induced ferroptosis? Whether its over-expression affect ferroptosis?

We thank the reviewer for this insightful comment. P0 cells have very low expression of HMGCR and are not significantly affected by the HMGCR-mediated mevalonate pathway, which is related to ferroptosis resistance. As suggested, we evaluated the effect of HMGCR overexpression in P0 cells. Overexpression of HMGCR rendered P0 cells resistant to RSL3-induced ferroptosis, as evidenced by a reduced percentage of 7AAD⁺ tumor cells and decreased lipid ROS production (Revised

Supplementary Fig 9a-9c). These findings further reinforce our hypothesis that HMGCR upregulation is a key driver of ferroptosis resistance in immune-refractory tumor cells. We have incorporated this data and additional discussion into the Results section and Figure legend of the revised manuscript as recommended.

[On Supplementary Fig. 9a-9c of the revised manuscript]

Supplementary Figure 9

[On page 08, Results section of the revised manuscript]

Given the crucial role of HMGCR upregulation in the ferroptosis-resistant phenotype of P3 cells, we examined the effect of HMGCR overexpression in P0 cells. Notably, HMGCR overexpression alone was sufficient to confer resistance to RSL3-induced ferroptosis, as indicated by a reduced proportion of 7-AAD⁺ cells and decreased lipid ROS levels (Supplementary Fig. 9a-9c). Together, these data indicate that CTL-mediated immune selection drives ferroptosis resistance through upregulation of HMGCR.

3. The gating strategy of flow cytometry analysis for in vivo tumor cell death (Fig. 2c), lipid ROS (Fig. 2d) and T cell number (Fig. 2e) and cytokine production (Fig. 2f) are needed.

As suggested, we have provided a detailed gating strategy for in vivo tumor cell death, lipid ROS, T cell number and cytokine production in the revised Supplementary Fig 21.

[On Supplementary Fig. 21 of the revised manuscript]

Supplementary Figure 21

in vivo tumor cell death (Fig. 2c)

lipid ROS (Fig. 2d)

T cell number (Fig. 2e)

cytokine production (Fig. 2f)

4. In the analysis shown in Fig.2e and 2f, why gp100 is used? In this case, the specific CTL recognize gp100 peptides should be detected.

The reviewer raises an important point regarding the use of gp100 in Fig 2e and 2f. In our study, we used the B16 melanoma model, where gp100 (glycoprotein 100) is a well-characterized tumor-associated antigen (TAA) naturally expressed by B16 tumor cells and widely used to monitor tumor-specific CD8⁺ T cell responses (Overwijk et al., J Exp Med, 2003). To assess the functional status of tumor-infiltrating T cells, we stimulated them with ex vivo gp100 peptides, which allowed us to evaluate antigen-specific CTL responses. In Fig 2f, cytokine production was measured following stimulation of tumor-infiltrating lymphocytes (TILs) with gp100 peptides, ensuring that the response reflected tumor antigen-specific CTLs rather than non-specific bystander T cells. We have clarified this rationale in the revised manuscript.

[On page 07, Results section of the revised manuscript]

Furthermore, **to confirm tumor antigen-specific T cell responses rather than non-specific T cells activation**, we stimulated tumor-derived single cells with either DMSO or gp100, a B16 tumor-associated antigen, and assessed IFN γ -producing CD8⁺ T cells.

5. What is the mechanism of E2F1 upregulation in P3 cells? Whether it is induced by in vitro CTL coculture or CTL supernatant?

The reviewer raised an important point regarding upregulation of E2F1 in immune-refractory tumors. In our previous studies (Noh KH et al., J Clin Invest. 2012), we demonstrated that CTL-mediated *in vitro* immune selection promotes the emergence of cancer cells with enhanced survival advantages, leading to the enrichment of immune-refractory cancer cells characterized by unique features, including cancer stem cell (CSC)-like properties. During this process, E2F1 contributes to CSC-like properties by activating FGFR signaling or inducing the stemness factor NANOG, and is further amplified through a reciprocal regulatory loop in which NANOG-driven AKT hyperactivation impairs RB function, thereby enhancing E2F1 activity (Song KH et al., Oncogenesis. 2017 and Oh SJ et al., Cancer Res. 2018). Based on this, we hypothesized that the upregulation of E2F1 could be a consequence of immune selection pressure induced by ICB or ACT therapies. To test this, we analyzed E2F1 expression across successive rounds of immune selection (P0 to P3) in TC-1 cells and A375 cells subjected to ICB and ACT, respectively. We observed a stepwise increase in E2F1 expression from P0 to P3 cells (revised Supplementary Fig. 14), suggesting that the elevated levels in P3 cells are likely the result from the enrichment of E2F1⁺ tumor cells during the immune selection process. We have included this information in the revised Results and Discussion section

[On Supplementary Fig. 14 of the revised manuscript]

Supplementary Figure 14. Son et al

[On page 10, Results section of the revised manuscript]

We next investigated the underlying mechanism responsible for the upregulation of E2F1 during immune editing. Our previous study showed that CTL-mediated immune selection enriches immune-refractory cancer cells with cancer stem cell (CSC)-like properties^{9,42}. In this regard, E2F1 contributes to CSC-like properties by activating FGFR signaling or inducing the stemness factor NANOG, and is further amplified through a reciprocal regulatory loop in which NANOG-driven AKT hyperactivation impairs RB function, thereby enhancing E2F1 activity^{42,43}. Based on this, we hypothesized that E2F1 upregulation may result from immune selection pressure induced by ICB or ACT therapies. To investigate this, we evaluated E2F1 expression in TC-1 and A375 cells across successive rounds of immune selection (P0 to P3) using ICB and ACT, respectively. We observed a stepwise increase in E2F1 expression from P0 to P3 cells (Supplementary Fig. 14), suggesting that the elevated E2F1 levels in P3 cells likely result from the enrichment of E2F1⁺ cells during the immune selection process. Collectively, our findings suggest that immune pressure induced by immunotherapy drives the selection of E2F1⁺ tumor cells, which subsequently upregulate HMGCRCR expression and contribute to ferroptosis resistance.

[On page 15, Discussion section of the revised manuscript]

Interestingly, the E2F1-binding element in the HMGCRCR promoter region is evolutionally conserved, suggesting the important role of E2F1 in HMGCRCR regulation. Indeed, compared to their parental P0 cells, E2F1 is upregulated in immune-edited P3 cells, and knockdown of E2F1 in P3 cells

reduces HMGCR levels and increases their susceptibility to ferroptosis. While we have established that the E2F1-HMGCR axis drives ferroptosis resistance, this raises an important question: how is E2F1 upregulated during cancer immunoediting? In this regard, our previous studies demonstrated that CTL-mediated immune selection enriches immune-refractory cancer cells with CSC-like properties, driven by E2F1 through FGFR signaling or NANOG induction. NANOG further amplifies E2F1 activity via AKT-mediated RB suppression, forming a reciprocal loop^{42,43}. In light of this, our current study suggest that immune pressure imposed by immunotherapy drives the selection of E2F1⁺ tumor cells, which in turn upregulate HMGCR expression and promotes ferroptosis resistance.

6. The evidence showing the clinical relevance of the E2F1-HMGCR axis to ICB response is too weak. The IHC staining of HMGCR or E2F1 on tumor tissue slides is suggested. Or incorporate this part into Fig. 4 or Fig. 6.

We appreciate the reviewer's comment. While we fully acknowledge the value of IHC staining for E2F1 and HMGCR in tumor tissues, this was not feasible due to the lack of matched tissue slides for the RNA-seq datasets used—an acknowledged limitation of our study. Nevertheless, we agree that such analysis will be important in future studies to validate and extend our preclinical findings. Accordingly, we have added a new paragraph in the revised Discussion section to address the study's limitations, including the small number of patient samples.

[On page 18, Discussion section of the revised manuscript]

Study limitations in our work include the small number of clinical samples, which restricted our ability to fully investigate the relationship among the E2F1-HMGCR axis, ferroptosis resistance, and ICB response in patient tumors. Our study provides mechanistic insight into how T cell-based immunotherapies, such as PD-1 blockade and adoptive T cell transfer, promote ferroptosis resistance through immunoediting in tumor cells. By identifying the E2F1-HMGCR axis as a key driver of this resistance, we demonstrate that HMGCR inhibition restores ferroptosis sensitivity, rendering immune-refractory tumor cells susceptible to ACT and PD-1 blockade.

However, due to the limited clinical samples, we were unable to directly assess ferroptosis-related features, such as lipid peroxidation, in patient specimens. Future studies involving larger patient cohorts and immunohistochemical analysis of ferroptosis markers will be essential to validate the clinical relevance of our findings. In addition, single-cell transcriptomic profiling of immune-refractory tumor models and patient samples will provide high-resolution insights into cell-type-specific HMGCR functions and regulatory networks. This approach is expected to refine our understanding of ferroptosis resistance and reveal complex intercellular interactions within immune-refractory tumors, ultimately informing precision immunotherapeutic strategies. Nonetheless, our findings provide a strong preclinical foundation for targeting ferroptosis resistance to improve the efficacy of T cell-based cancer immunotherapies.

7. Since simvastatin can promote ferroptosis, whether it can enhance tumor cell death in the co-culture with CTL?

We appreciate the reviewer's insightful comment regarding whether simvastatin can enhance tumor cell death in co-culture with CTLs. As response in question #1, in direct CTL-tumor cell co-culture systems, the rapid onset of apoptosis compromises the ability to detect temporally delayed ferroptotic processes. Therefore, we performed co-culture experiments using P3 tumor cells with CTL-derived supernatants in the presence or absence of simvastatin. Simvastatin treatment significantly enhanced tumor cell death under CTL supernatant exposure, and increased lipid ROS accumulation, indicative of ferroptosis (revised supplementary figure 15). We have incorporated these findings into the Results section and Figure legend of the revised manuscript as recommended. We sincerely appreciate the reviewer's valuable feedback, which has helped refine our study and strengthen our conclusions. We have included this information in the revised Results section

[On Supplementary Fig. 15 of the revised manuscript]

Supplementary Figure 15.

[On page 11, Results section of the revised manuscript]

Notably, simvastatin synergized with RSL-3 in immune-refractory P3 cells but not in parental P0 cells (Fig 6a and 6b). Consistent with our findings from genetic inhibition of HMGCR, simvastatin treatment enhanced ferroptosis in immune-refractory P3 cells induced by either RSL3 or CTL-derived supernatant, as evidenced by increased levels of 7-AAD+ tumor cells and lipid ROS (Fig. 6c-6d and Supplementary Fig. 15).

8. Do the results shown in Supplementary Fig. 3a-b suggest that T-cell supernatant can directly induce ferroptosis of A375 P0 cells? How to explain this?

We appreciate the reviewer's comment regarding whether T-cell supernatant can directly induce ferroptosis. IFN γ secreted by activated T cells has been reported to induce ferroptosis, thereby contributing to the anti-tumor efficacy of immunotherapy (Weimin Wang et al., Nature, 2019). Mechanistically, IFN γ -mediated STAT1 activation has been shown to regulate lipid metabolism and

oxidative stress pathways that promote ferroptosis, including the downregulation of SLC7A11 and SLC3A2 and upregulation of ACSL4-dependent lipid peroxidation. In addition, activated NY-ESO-1-specific T cells have been shown to secrete IFN γ upon recognizing NY-ESO-1-positive melanoma cell lines, including A375 cells (Jean Francois Fonteneau et al., J Immunol., 2016). In this context, we observed that supernatant from activated NY-ESO-1-specific T cells increased ferroptosis in A375 P0 cells. Notably, these effects were abolished by IFN γ -neutralizing antibodies or Lip-1 treatment (Fig. 3h and revised Supplementary Fig. 10a, 10b), suggesting a critical role for soluble IFN γ in CTL-induced ferroptosis. These results further support the concept that IFN γ released by antigen-specific T cells can directly modulate ferroptosis-related metabolic vulnerability in tumor cells. However, while IFN γ -neutralizing antibodies reversed the majority of the enhanced ferroptosis, a minor residual effect remained, suggesting that additional factors besides IFN γ may contribute to ferroptosis induction. This observation highlights the need for further investigation to elucidate these mechanisms. We have included this information in the revised Results section

[On page 11, Results section of the revised manuscript]

Consistent with our findings from genetic inhibition of HMGCR, simvastatin treatment enhanced ferroptosis in immune-refractory P3 cells induced by either RSL3 or CTL-derived supernatant, as evidenced by increased levels of 7-AAD⁺ tumor cells and lipid ROS (Fig. 6c-6d and Supplementary Fig. 15). These effects were abolished by IFN γ -neutralizing antibodies or Lip-1, supporting a critical role for soluble IFN γ in CTL-induced ferroptosis. This aligns with previous findings that IFN γ from antigen-specific T cells promotes ferroptosis via STAT1-mediated downregulation of SLC7A11/SLC3A2 and enhanced lipid peroxidation¹⁴. However, as IFN γ blockade substantially reversed ferroptosis, a residual effect persisted, suggesting the involvement of other factors involved. Since we observed that E2F1 overexpression in A375 P0 cells phenocopied ferroptosis-resistance of A375 P3 cells (Fig. 4l and 4m), we then asked whether simvastatin could reverse the ferroptosis resistant properties of E2F1 overexpressed tumor cells.

Minor points:

1. The cell death percentage shown in Fig. 4L is too low, suggesting no significant cell death was induced. This experiment needs to be repeated with higher RSL3 or longer treatment time.

We thank the reviewer for the comment. As suggested, we have repeated the experiment and replaced the data image in Fig. 4l of revised manuscript.

[On Fig. 4l of the revised manuscript]

Fig. 4

2. The band of HMGCRCR shown in Supplementary Fig. 2a seems modified, please show the original exposure band of HMGCRCR

We thank the reviewer for raising this concern. We have replaced the HMGCRCR blots of original Supplementary Fig. 2a (now revised Supplementary Fig. 4a) as recommended.

[On Supplementary Fig. 4a of the revised manuscript]

Supplementary Figure 4

Dear Reviewer,

My coauthors and I appreciate the detailed and relevant comments that have been raised in response to our manuscript entitled “**The E2F1-HMGCR axis promotes ferroptosis resistance in immune refractory tumor cells**”. Your assistance has been invaluable and has helped us improve the quality of our manuscript. We have fully addressed your comments and amended the manuscript content accordingly. We have also provided point-by-point responses to the critiques. The reviewer questions are in **bold text** while our responses are in regular or underlined text (if necessary). In the revised manuscript, the changes in the revised manuscript are marked in **RED**. Once again, we thank you for your time, consideration, and invaluable guidance.

Responses to reviewer:

Reviewer #2

The authors are appreciated for providing more experimental evidence that substantiate their hypothesis, and formally acknowledged and discussed the limitation of the current study. They have addressed most of my concerns. I still have two questions that need attention from the authors:

1) In response to my question 3 "it is still not answered that how E2F1 was upregulated from P0 to P3?". The authors proposed that immune killing pressure selected E2F1 high cancer cells. Do they think the E2F1 expression levels are heterogeneous among different cancer cell clones before immune selection, or E2F1 expression can be inducible during immune selection? Please be more specific.

We appreciate reviewer's positive assessment and constructive feedback. We have incorporated the suggested revisions into the Results and Discussion section of the revised manuscript.

[On page 10, Results section of the revised manuscript]

We observed a stepwise increase in both E2F1 expression and the proportion of E2F1⁺ cells from P0 to P3 cells (Supplementary Fig. 14), suggesting that E2F1 upregulation in P3 tumor cells results from the immune pressure-driven enrichment of pre-existing E2F1⁺ tumor cells during immunotherapy. Collectively, our findings suggest that immune pressure induced by immunotherapy drives the selection of E2F1⁺ tumor cells, which subsequently upregulate HMGCRC expression and contribute to ferroptosis resistance.

[On page 16, Discussion section of the revised manuscript]

In light of this, our current study suggests that immune pressure imposed by immunotherapy drives the selective enrichment of pre-existing E2F1⁺ tumor cells within the parental population, which in turn upregulate HMGCRC expression and promote ferroptosis resistance.

2) If both apoptosis and ferroptosis contribute to immune killing of cancer cells, and E2F1 happens to regulate both cell death pathways, the authors need to strengthen the novelty of their findings in the context of cell death induced by T cells reinvigorated by anti-PD1 therapy.

We appreciate the reviewer's valuable suggestion. We have revised the Discussion section to better emphasize the novel role of E2F1 in regulating apoptosis and ferroptosis in the context of CTL-mediated cytotoxicity enhanced by anti-PD-1 therapy.

[On page 16, Discussion section of the revised manuscript]

In this regard, our previous studies demonstrated that CTL-mediated immune selection enriches immune-refractory cancer cells with CSC-like properties and resistance to CTL-mediated apoptosis, driven by E2F1 through FGFR signaling or NANOG induction. NANOG further amplifies E2F1 activity via AKT-mediated RB suppression, forming a reciprocal loop^{42,43}. In light of this, our current study suggests that immune pressure imposed by immunotherapy drives the selective enrichment of pre-existing E2F1⁺ tumor cells within the parental population, which in turn upregulate HMGCRC expression and promote ferroptosis resistance. Building upon previous findings, we further establish E2F1 as a key regulator of both apoptosis and ferroptosis resistance in the context of tumor cell death mediated by CTLs reinvigorated through anti-PD1 therapy.